# Learning to Route: Per-Sample Adaptive Routing for Multimodal Multitask Prediction

**Marzieh Ajirak**[1][*]   **Oded Bein**[1]   **Ellen Rose Bowen**[1]   **Dora Kanellopoulos**[1]
**Avital Falk**[1]   **Faith Gunning**[1]   **Nili Solomonov**[1][†]   **Logan Grosenick**[1,2][†]
[1]Weill Cornell Medicine, Cornell University, NY, USA
[2]Feil Family Brain & Mind Research Institute, Cornell University, NY, USA

## Abstract

We propose a unified framework for adaptive routing in multitask, multimodal prediction settings where data heterogeneity and task interactions vary across samples. We introduce a routing-based architecture that dynamically selects modality processing pathways and task-sharing strategies on a per-sample basis. Our model defines multiple modality paths, including raw and fused representations of text and numeric features, and learns to route each input through the most informative modality-task expert combination. Task-specific predictions are produced by shared or independent heads depending on the routing decision, and the entire system is trained end-to-end. We evaluate the model on both synthetic data and real-world psychotherapy notes, predicting depression and anxiety outcomes. Our experiments show that our method consistently outperforms fixed multitask or single-task baselines, and that the learned routing policy provides interpretable insights into modality relevance and task structure. This addresses critical challenges in personalized healthcare by providing per-subject adaptive information processing that accounts for data and task correlation heterogeneity.

## 1   Introduction

Modern predictive models increasingly operate in settings with multiple heterogeneous input modalities and correlated outputs. In domains such as clinical informatics and behavioral health (Preis et al., 2022), data arrive in diverse formats that include structured numerical features (clinical scales, sensor measurements) and unstructured text (clinician notes, patient narratives) (Baltrušaitis et al., 2018; Rajkomar et al., 2018). These modalities differ in structure, coverage, semantic density, and sample-level informativeness. On the other hand, multiple predictive targets (for example, depression and anxiety scores) often correlate but do not overlap entirely. This combination motivates multimodal multitask learning models that integrate heterogeneous inputs while modeling structured relationships across tasks.

Multimodal learning aims to learn joint representations from diverse data sources. However, most existing approaches assume fixed fusion strategies and complete modality availability (Ruder, 2017; Liu et al., 2022). Similarly, multitask learning (MTL) typically relies on globally shared architectures, applying the same parameter sharing scheme to all inputs. However, these assumptions are often violated in real-world settings. Modality quality and informativeness can vary substantially across samples, and task relationships may differ depending on latent factors such as individual behavior, context, or data completeness. Ignoring these forms of heterogeneity leads to suboptimal representations and reduced predictive performance and generalization.

---

[*]Corresponding authors: Marzieh Ajirak, maa4083@med.cornell.edu; Logan Grosenick, log4002@med.cornell.edu

[†]Joint senior authors

39th Conference on Neural Information Processing Systems (NeurIPS 2025).

To address these limitations, we propose a unified framework that performs adaptive expert routing over both modality and task configurations (Shazeer et al., 2017; Ma et al., 2018; Rosenbaum et al., 2017). Our model defines a set of modality transformation paths, including unimodal and fused representations, and a set of prediction heads corresponding to single-task and multitask supervision structures. A learnable routing mechanism selects a personalized expert pathway for each input by modeling a probabilistic mixture over modality-task combinations, capturing both input-dependent modality preferences and task-level interaction structure. This approach generalizes conditional computation and mixture-of-experts (MoE) frameworks (Liu et al., 2024a) to settings where latent structure exists across inputs, supervision targets, and representational hierarchies.

The routing policy is parameterized by neural networks and trained jointly with all expert modules using backpropagation. To promote policy diversity and mitigate expert collapse (where a subset of experts get nearly all the traffic), we incorporate an entropy regularization term (Fedus et al., 2022). This design enables sample-specific selection of both data representations and task decoders, effectively adapting the computation graph to underlying data geometry and supervision structure.

We validate the proposed framework through a series of experiments on synthetic and real-world data. In synthetic data with controlled variation in modality relevance and task correlation, our model outperforms fixed multitask and single-task baselines while recovering interpretable routing policies. In real-world psychotherapy data with structured assessments and unstructured clinician notes (Benton et al., 2017; Niu et al., 2024), the model improves prediction of anxiety and depression outcomes and reveals routing decisions that align with intuitive task-modality interactions. These results demonstrate that adaptive routing over representation and supervision structure is a powerful mechanism for modeling heterogeneous, multimodal prediction tasks.

In summary, our **major contributions** are: (1) we develop a modular architecture supporting multiple modality transformation paths and adaptive task-sharing schemes, (2) we design a probabilistic routing mechanism that dynamically selects, for each input, both optimal modality pathways and task configurations based on input and output structure, and (3) we demonstrate significant improvements in both prediction accuracy and interpretability across synthetic and real-world clinical datasets. Our approach consistently outperforms standard multitask and single-task baselines, with immediate potential applications for enhancing decision support in mental healthcare and broader medical contexts. The framework's ability to adapt to heterogeneous data while modeling structured relationships across tasks makes it particularly valuable for real-world clinical environments where data heterogeneity and quality vary (across patients, clinicians, sites, etc.) and outcome measures are often interdependent (e.g., multiple clinical scales or physiological measurements). Open source code is available at: `https://github.com/Grosenick-Lab-Cornell/learning-to-route`.

## 2 Related work

### 2.1 Multimodal learning in clinical and mental health contexts

A growing body of research investigates multimodal learning with structured clinical data (e.g., electronic health records, standardized assessments) and unstructured text (e.g., clinician notes) to improve outcome prediction. In general medical AI, combining tabular EHR features with narrative notes has led to measurable gains in predictive performance (Lyu et al., 2022). In the mental health domain, fusion of structured and unstructured data has yielded similar benefits. For instance, Garriga et al. (2023) predicted 28-day psychiatric crisis risk using both structured EHR variables and clinical note text, reporting that models leveraging both modalities outperformed unimodal baselines. Other studies have found that incorporating text embeddings derived from models like BERT into structured-input pipelines improves accuracy across various clinical tasks (Ye et al., 2024).

However, the gains from multimodal fusion are not consistent across settings. For example, Kotula et al. (2025) found that augmenting vital signs and lab values with concept-extracted notes led to only marginal improvements for ICU deterioration prediction. These mixed results suggest that the informativeness of unstructured text and the chosen fusion strategy can critically impact performance. Our approach addresses this limitation by supporting adaptive fusion (Sahu and Vechtomova, 2021; Xue and Marculescu, 2023; Ajirak et al., 2023b). Instead of using a fixed integration schema, the model determines how to combine structured and unstructured inputs separately for each sample. This enables more flexible use of the available modalities, depending on their informativeness for the individual case.

## 2.2 Multitask learning for mental health prediction

Multitask learning (MTL) (Kendall et al., 2018) offers a principled approach for modeling multiple correlated clinical outcomes. Conditions such as major depressive disorder and generalized anxiety frequently co-occur and exhibit overlapping symptom profiles, yet clinical models often treat them as independent targets or apply joint statistical models that fail to capture their structured dependencies. MTL enables parameter sharing across tasks, which supports the learning of shared representations that can improve generalization and predictive performance. Prior work has demonstrated the effectiveness of this approach in mental health settings. For example, Buddhitha and Inkpen (2023) constructed a shared encoder with both hard and soft parameter sharing to jointly model mental illness and suicide ideation risk, outperforming task-specific baselines. In another study, Saylam and İncel (2024) showed that jointly predicting depression, anxiety, and stress levels led to performance gains for depression and stress compared to single-task models. These findings show that MTL architectures with shared encoders and task-specific decoders can leverage cross-task structure while preserving task specialization.

Task-relatedness plays a central role in the effectiveness of multitask learning. When tasks are closely correlated or reflect underlying causal relationships, multitask models often achieve positive transfer (Standley et al., 2020). In contrast, unrelated tasks can lead to negative transfer and representation interference (Zhang and Yang, 2021). To address this, recent work has introduced methods that learn task relationships directly from data. The Multi-gate Mixture-of-Experts (MMoE) architecture, for example, assigns each task a dedicated gating network that selects among a shared pool of experts, thereby enabling flexible combinations of shared and task-specific computation (Ma et al., 2018). These designs relax the binary distinction between fully shared and fully independent decoders and support a continuum of sharing that adapts to task similarity (Ruder, 2017). In the context of mental health prediction, where tasks often reflect partially overlapping symptom dimensions, such flexibility is especially important. We build on this idea by introducing an adaptive routing mechanism that integrates with MTL and selects the appropriate degree of sharing across tasks based on sample-level signals.

## 2.3 Adaptive routing and mixture-of-experts for heterogeneous inputs/tasks

Recent advances in adaptive routing (Rosenbaum et al., 2017) and Mixture-of-Experts (MoE) (Jacobs et al., 1991) architectures offer flexible strategies for modeling heterogeneity across both inputs and tasks. MoE models consist of multiple specialized subnetworks ("experts") and a trainable router that assigns each input to a subset of these experts based on their characteristics (Mu and Lin, 2025; Liu et al., 2024b). This setup enables conditional computation, where different experts can focus on different regions of the input space or specialize in particular tasks. In large-scale language and vision models, sparsely activated MoEs have improved computational efficiency and generalization compared to densely connected alternatives (Zhou et al., 2022).

In clinical applications, conditional routing offers practical benefits for handling input variability. For instance, a model can assign a sample with detailed textual notes to a text-specialized expert, while directing a sample with only numerical features to a numerical-specialized expert. This flexibility supports personalized prediction pipelines without requiring separate models for every data configuration. Recent work in medical machine learning has applied these principles to multimodal tasks. The dynamic routing framework proposed by Wu et al. (2025) selects modality-task combinations on a per-sample basis, capturing the dependencies between clinical outcomes and input modalities. Their model learns a modality fusion strategy using mutual information regularization, which guides the decomposition of each sample's data into shared and distinct components.

Routing mechanisms have also been applied to task-level adaptation. For example, Rosenbaum et al. (2017) introduced routing networks that learn input-dependent paths through modular function blocks. This architecture allows the model to activate shared components when beneficial and to fall back on task-specific routes when task interference arises. These methods collectively demonstrate the value of flexible routing schemes in domains with complex, variable inputs and overlapping prediction objectives. Our work builds on this foundation by integrating sample-level routing over both modalities and tasks, enabling a unified framework for adaptive multimodal multitask learning.

# 3 Methodology

## 3.1 Bidirectional Transformation Between Structured and Unstructured Modalities

Our approach addresses the multimodal nature of psychotherapy datasets by introducing flexible, bidirectional transformations between structured numerical data and unstructured text. Rather than treating each modality as isolated, we design a unified architecture capable of transforming and integrating representations across formats. This enables the model to operate uniformly across patients with different modality availability, data missingness, or data quality.

Let $X_{\text{num}} \in \mathbb{R}^{d_{\text{num}}}$ represent the structured numerical input, and $X_{\text{text}} \in \mathcal{T}$ denote unstructured textual data. We define two learned transformation functions:

$$f_{\text{num2text}} : \mathbb{R}^{d_{\text{num}}} \to \mathcal{T}, \tag{1}$$

$$f_{\text{text2num}} : \mathcal{T} \to \mathbb{R}^{d_{\text{text}}}, \tag{2}$$

where $f_{\text{num2text}}$ converts numerical input into semantically meaningful natural language, and $f_{\text{text2num}}$ encodes text into numerical embeddings. Here, $f_{\text{text2num}}$ can be implemented using sentence embedding models such as MPNet (Song et al., 2020) or Sentence-BERT (Reimers and Gurevych, 2019). The generated embedding is a structured numerical representation that can be integrated with traditional statistical models or deep learning architectures designed for numerical inputs.

We define the following four cases:

1. **Text-only (T1):** Use only $X_{\text{text}}$. Numerical input is ignored or unavailable.

2. **Numerical-only (N1):** Use only $X_{\text{num}}$. Text input is ignored or unavailable.

3. **Textualized Numerical + Text (T2):** Apply $f_{\text{num2text}}(X_{\text{num}})$ to produce $X_{\text{num}}^{(\text{text})}$, and concatenate with original text:

$$X = \text{concat}_{\text{text}}(X_{\text{num}}^{(\text{text})}, X_{\text{text}}), \tag{3}$$

   forming a unified textual input passed to a text-native model (e.g., transformer). This allows us to employ pre-trained language models such as BERT (Devlin, 2018) and MPNet (Song et al., 2020) to process text data.

4. **Numerical + Text Embedding (N2):** Apply $f_{\text{text2num}}(X_{\text{text}})$ to produce $X_{\text{text}}^{(\text{num})}$, and concatenate with original numerical features:

$$X = \text{concat}_{\text{num}}(X_{\text{text}}^{(\text{num})}, X_{\text{num}}), \tag{4}$$

   resulting in a unified numerical representation passed to a numerical backbone.

These paradigms enable flexible fusion strategies that are adaptable to data quality, availability, and downstream model compatibility. Figure 4 illustrates these conversions and concatenation. The two-way conversion creates a flexible bridge between data types. It lets structured data be understood in natural language terms (textualization) while also transforming text into numerical formats that work with traditional data models.

## 3.2 Multitask vs. Single-Task Learning Objectives

We adopt a multitask learning (MTL) framework to address correlated clinical outcomes (depression, anxiety) of psychotherapy (Ruder, 2019). The performance of commonly used multitask models often depends on the relationships between tasks. Therefore, studying the trade-offs between task-specific objectives and inter-task dependencies is crucial. We focus on predicting two common key clinical outcomes: Depression (measured with the Patient Health Questionnaire-9 [PHQ-9] (Kroenke et al., 2001)) and anxiety (Generalized Anxiety Disorder-7 [GAD7] (Spitzer et al., 2006)). These measures are correlated and so can provide both distinct and redundant information about a patient's symptoms. Our approach therefore, supports both single-task learning (STL) and multitask learning (MTL):

1. **Single-Task Learning (STL):** Trains two independent models, one per outcome. Each model $f_k(X^{(i)})$ predicts both the outcome $\hat{y}_k$ and its log-variance $\log \sigma_k^2$, where

$k \in \{\text{PHQ}, \text{GAD}\}$ and $X^{(i)}$ is the modality-transformed input for paradigm $i \in \{\text{T1}, \text{N1}, \text{T2}, \text{N2}\}$. For each task:

$$f_k(X^{(i)}) = \left( \hat{y}_k(X^{(i)}), \log \sigma_k^2(X^{(i)}) \right), \tag{5}$$

the heteroscedastic loss for task $k$ is:

$$\mathcal{L}_{\text{STL},k}(X^{(i)}, y_k) = \frac{1}{2} \cdot \frac{(y_k - \hat{y}_k(X^{(i)}))^2}{\sigma_k^2(X^{(i)})} + \frac{1}{2} \log \sigma_k^2(X^{(i)}), \tag{6}$$

and the total STL loss is:

$$\mathcal{L}_{\text{STL}} = \mathcal{L}_{\text{STL,PHQ}} + \mathcal{L}_{\text{STL,GAD}}. \tag{7}$$

2. **Multi-Task Learning (MTL):** Uses a shared encoder followed by task-specific heads. The shared encoder produces a latent representation $z = f_{\text{shared}}(X^{(i)})$.

    Each task head then outputs both a mean prediction and log-variance:

$$f_k(z) = \left( \hat{y}_k(z), \log \sigma_k^2(z) \right), \quad k \in \{\text{PHQ}, \text{GAD}\}, \tag{8}$$

with the heteroscedastic loss per task being:

$$\mathcal{L}_{\text{MTL},k}(z, y_k) = \frac{1}{2} \cdot \frac{(y_k - \hat{y}_k(z))^2}{\sigma_k^2(z)} + \frac{1}{2} \log \sigma_k^2(z), \tag{9}$$

and the total MTL loss is:

$$\mathcal{L}_{\text{MTL}} = \sum_{k \in \{\text{PHQ,GAD}\}} \mathcal{L}_{\text{MTL},k}(z, y_k). \tag{10}$$

MTL improves generalization when tasks share underlying signals, while STL is preferred if task-specific features dominate or tasks conflict.

## 3.3 Modeling sample heterogeneity via probabilistic expert routing

Psychotherapy data includes inherent heterogeneity. First, there is variation across patients due to both individual heterogeneity in patient biology and symptom presentation, as well as due to data modality, missingness, and quality per patient, clinician, or site (Preis et al., 2023; Ajirak et al., 2023a). Second, there is variability in the extent to which measured outcomes correlate. For example, some patients may present with both depression and anxiety symptoms that change in tandem during treatment, while others may experience specific changes to just anxiety or depression symptoms. Rather than applying a fixed modality paradigm or learning strategy across all samples, we assume samples vary in modality informativeness and task relevance. To model this heterogeneity, we introduce a hierarchical mixture-of-experts architecture that probabilistically routes each sample to one of eight expert paths: $\{\text{T1}, \text{T2}, \text{N1}, \text{N2}\} \times \{\text{STL}, \text{MTL}\}$.

**Routing Architecture:** We define a two-stage probabilistic routing mechanism that dynamically selects among modality paths and task setups on a per-sample basis.

- **Modality Router:** Given $X^{(i)}$, a gating function $r_{\text{mod}}$ outputs a probability distribution over the four modality paths (T1, T2, N1, N2):

$$\pi_{\text{mod}} = \text{softmax}(r_{\text{mod}}(X_{\text{num}}, X_{\text{text}})) \in \mathbb{R}^4. \tag{11}$$

    Each modality path $i \in \{1, 2, 3, 4\}$ corresponds to a specific transformation of the input, resulting in a modality-specific representation $X^{(i)}$.

- **Task Router:** For each modality-transformed input $X^{(i)}$, a second gating function $r_{\text{task}}$ computes a distribution over task strategies (STL or MTL):

$$\pi_{\text{task}}^{(i,j)} = \text{softmax}(r_{\text{task}}(X^{(i)})) \in \mathbb{R}^2. \tag{12}$$

    Here, $j \in \{1, 2\}$ indexes STL and MTL, respectively.

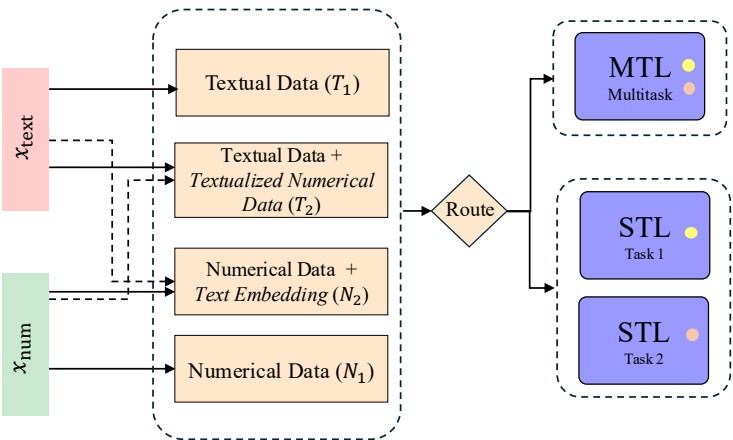

Figure 1: Overview of the adaptive multimodal prediction framework. Each data sample, made up of structured numerical features and unstructured clinical notes, passes through a probabilistic router that directs it to the most suitable expert. Depending on the input, the system routes samples to either single-task (STL) or multitask (MTL) models designed for numeric, text, or mixed modalities. This Mixture-of-Experts setup adjusts to the variability in the data and helps improve overall prediction performance.

Each combination of modality path $i$ and task strategy $j$ defines an expert model $f^{(i,j)}$ that outputs heteroscedastic predictions for both outcomes:

$$f^{(i,j)}(X^{(i)}) = \left( \hat{y}_{\text{PHQ}}^{(i,j)}, \log \sigma_{\text{PHQ}}^{2(i,j)}, \hat{y}_{\text{GAD}}^{(i,j)}, \log \sigma_{\text{GAD}}^{2(i,j)} \right).$$

The overall predictive outputs are computed as a mixture over expert paths:

$$\hat{y}_{\text{PHQ}} = \sum_{i=1}^{4} \sum_{j=1}^{2} \pi_{\text{mod}}^{(i)} \cdot \pi_{\text{task}}^{(i,j)} \cdot \hat{y}_{\text{PHQ}}^{(i,j)}, \tag{13}$$

$$\hat{y}_{\text{GAD}} = \sum_{i=1}^{4} \sum_{j=1}^{2} \pi_{\text{mod}}^{(i)} \cdot \pi_{\text{task}}^{(i,j)} \cdot \hat{y}_{\text{GAD}}^{(i,j)}. \tag{14}$$

Each expert also contributes to the total uncertainty-aware loss. For soft routing, we compute the expected loss across all expert paths:

$$\mathcal{L}_{\text{total}} = \sum_{k \in \{\text{PHQ},\text{GAD}\}} \sum_{i=1}^{4} \sum_{j=1}^{2} \pi_{\text{mod}}^{(i)} \cdot \pi_{\text{task}}^{(i,j)} \cdot \left[ \frac{1}{2} \cdot \frac{(y_k - \hat{y}_k^{(i,j)})^2}{\sigma_k^{2(i,j)}} + \frac{1}{2} \log \sigma_k^{2(i,j)} \right]. \tag{15}$$

In the case of hard routing, we replace the mixture with discrete selection and only one $(i, j)$ pair contributes to the prediction and loss. We explore both hard routing (differentiable approximation to the argmax operator) and soft routing (mixture), trained end-to-end using total task loss. In soft routing, gradients flow through all paths, encouraging specialization. In hard routing, selection is treated as discrete via Gumbel-Softmax reparameterization (Jang et al., 2016). This architecture captures complex heterogeneity in both data representation and outcome structure, automatically discovering which modality and learning scheme are best suited for each sample.

## 4 Experimental setup: synthetic multitask regression with adaptive routing

### 4.1 Synthetic data generation

To evaluate our model's ability to learn input-dependent routing policies, we construct a synthetic multitask, multimodal regression benchmark. Each sample consists of two modalities: a numeric

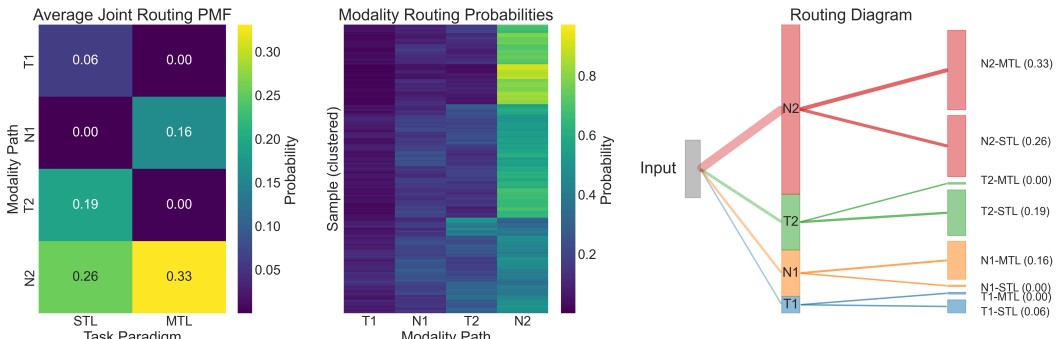

Figure 2: **Routing behavior visualizations.** (Left) *Average Joint Routing PMF:* Heatmap showing average routing probabilities across all eight (modality by task paradigm) expert combinations. The model assigns most mass to N2-STL and N2-MTL, indicating a preference for fused numerical+text representations, especially under MTL. (Middle) *Modality Routing Probabilities:* Clustered heatmap of sample-specific routing distributions over four modality paths. The model segments the input space into distinct routing patterns, reflecting input-dependent computation. (Right) *Routing Diagram:* Sankey plot illustrating average routing policy. Edge widths are proportional to route probabilities. The architecture dynamically allocates across experts based on input characteristics.

vector $x^{(\text{num})} \in \mathbb{R}^{d_{\text{num}}}$ and a textual vector $x^{(\text{text})} \in \mathbb{R}^{d_{\text{text}}}$. Both modalities contribute signal to both tasks, but their relevance varies across the input space.

To simulate this heterogeneity, we define input-dependent latent preferences over modality informativeness and task relevance. For each sample, we generate two outputs $y_1$ and $y_2$ corresponding to Task 1 and Task 2. These targets are nonlinear functions of both modalities, with task-specific coefficients and feature maps:

$$y_1 = \alpha_1^\top x^{(\text{num})} + \beta_1^\top \phi(x^{(\text{text})}) + \gamma_1 \cdot \sin(\omega_1^\top x^{(\text{num})}) + \epsilon_1,$$
$$y_2 = \alpha_2^\top x^{(\text{text})} + \beta_2^\top \psi(x^{(\text{num})}) + \gamma_2 \cdot \cos(\omega_2^\top x^{(\text{text})}) + \epsilon_2,$$

where $\phi$ and $\psi$ are random Fourier feature maps and $\epsilon_k \sim \mathcal{N}(0, \sigma^2)$ is Gaussian noise. The parameters $\alpha_k, \beta_k, \gamma_k, \omega_k$ are sampled independently for each trial. This setup introduces nonlinear, cross-modality interactions and supports fine-grained control over the input-task-modality relationships.

Taken together, Figures 2 and 3 illustrate the model's ability to align routing decisions with the latent structure of the task-modality landscape. Routes with higher selection probability correspond to lower predictive error, confirming that the learned policy not only adapts to input characteristics but also supports improved task performance. These results validate the effectiveness of probabilistic expert routing as a mechanism for uncovering and exploiting sample-specific patterns in multimodal multitask prediction.

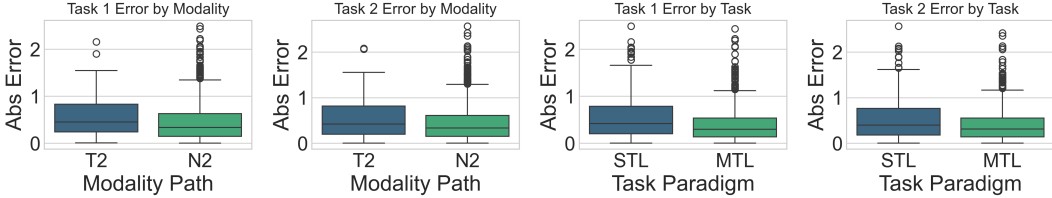

Figure 3: **Task-specific error distributions under hard routing.** Boxplots show absolute prediction errors for each task, stratified by the selected modality path (T2, N2) and task paradigm (STL, MTL). The T2 and N2 configurations yield the lowest errors overall and dominate the hard routing assignments. Notably, T1 and N1 routes are effectively pruned by the router, receiving near-zero selection probability. MTL performs slightly better than STL, but not decisively enough to eliminate STL paths. These results demonstrate the benefit of adaptive routing across modalities and tasks.

## 4.2 Real-world dataset: Mental Health in Healthcare Workers

We applied our approach to an augmented real-world psychotherapy dataset involving healthcare workers (hereafter "patients" for brevity) who presented with anxiety and/or depression symptoms during the course of the COVID-19 pandemic (Kanellopoulos et al., 2021; Solomonov et al., 2022). In this dataset, clinicians documented each session through unstructured notes (textual input, T1) and completed standardized clinical questionnaires (structured input, N1). Each patient was assessed using two outcome measures: depression severity (PHQ-9; Task 1) and anxiety severity (GAD-7; Task 2). To unify modalities, we transformed the structured numerical inputs into natural language using a fine-tuned text generation model, enabling early fusion with clinical notes to form a joint textual representation (T2). Figure 4 provides an example of this textualization process. Conversely, we encoded the unstructured clinical notes into numerical embeddings using a pretrained BERT model and concatenated these embeddings with the original structured features to construct a fused numerical representation (N2). This bidirectional transformation supports flexible modality routing and aligns heterogeneous inputs in a common representation. We evaluate using: (i) the predictive value of different modality paths, (ii) the benefit of multitask and heteroscedastic training objectives, and (iii) the effect of adaptive routing in selecting appropriate expert configurations.

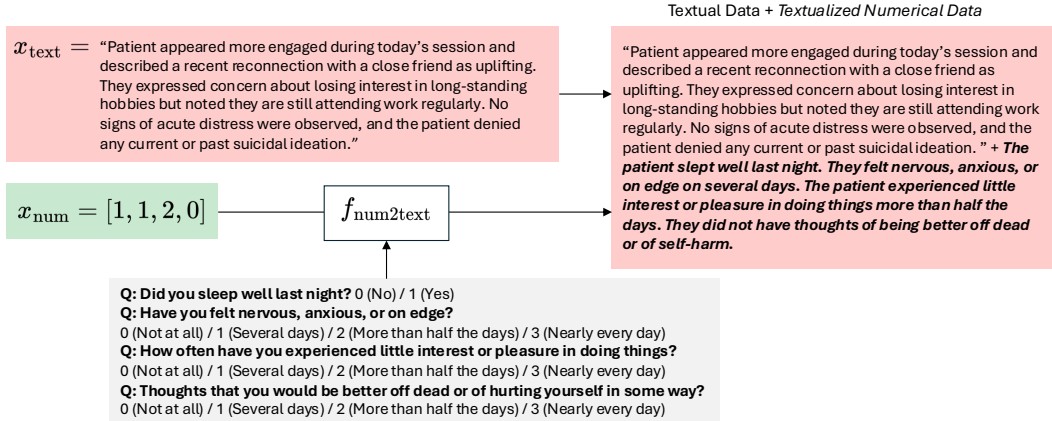

Figure 4: **Example of early fusion modality conversion.** (Left) Original data with numerical responses and therapist notes (not from a real patient to ensure privacy). (Right) Transformed representation where numerical responses are converted into text and concatenated with the original text, creating a unified textual modality.

Table 1 summarizes the six expert configurations used in our model, derived from combinations of modality inputs (text, numerical, or both) and training objectives (STL or MTL). These correspond to the eight possible (modality by task) pathways in our mixture-of-experts routing scheme. Each configuration supports different patient profiles (for instance, patients with only clinical notes are routed to text-only experts, while patients with multiple structured assessments and correlated outcomes benefit more from MTL).

Table 2 reports the predictive performance (RMSE) for each outcome. T1 (text-only) performs worst, particularly on PHQ-9, likely due to missing signal in free-text for structured outcomes. N1 (numerical-only) improves performance modestly. T2 (text + textualized numeric) achieves the best results overall, confirming that converting structured data into natural language and early-fusing it with notes enhances representation. N2 (numerical + embedded text) also performs well, but slightly under performs T2, suggesting limitations in treating text as static embeddings rather than contextual language input.

It also compares models trained with different objectives and routing strategies. STL under-performs across both outcomes, especially PHQ-9, where inter-task signal is stronger. MTL improves over STL by learning shared representations. Introducing learned routing improves both tasks, confirming that adaptive pathway selection benefits generalization. Finally, combining routing and heteroscedastic loss yields the best overall performance, demonstrating that routing complements uncertainty-aware weighting. This model dynamically selects both modality and task structure on a per-sample basis and adjusts supervision strength based on estimated noise.

Table 1: Overview of expert routes with their functions and ideal patient profiles.

| Route | Function | Interpretation |
|---|---|---|
| **Numerical (STL)** | Uses only structured scores (e.g., PHQ-9) to predict one outcome. | Patients with only standardized questionnaire scores (PHQ-9, GAD-7) and no, short, or noisy therapist notes. |
| **Numerical (MTL)** | Uses structured scores to predict multiple outcomes. | Patients with multiple structured scores available (PHQ-9, GAD-7) where outcomes are correlated. |
| **Text (STL)** | Uses only therapist notes to predict one outcome. | Patients without or with noisy structured assessments but with detailed clinical notes, with only one outcome available or two outcomes with possible low correlation. |
| **Text (MTL)** | Uses only therapist notes to predict multiple outcomes. | Patients whose therapist notes describe multiple mental health problems (e.g., mentions of both depression and anxiety) without or with noisy structured scores. |
| **Numerical + Text (STL)** | Uses both structured and unstructured data but predicts only one outcome at a time. | Patients with both questionnaire scores and therapist notes with low outcome correlation. |
| **Numerical + Text (MTL)** | Uses both structured and unstructured data to predict multiple outcomes jointly. | Patients with both structured scores and clinical text, where multiple conditions (e.g., depression & anxiety) need prediction. |

Table 2: RMSE on PHQ-9 / GAD-7 for all fixed-path baselines and model-level variants.

| Configuration | PHQ-9 RMSE | GAD-7 RMSE |
|---|---|---|
| *Modality-path baselines (STL)* | | |
| T1 (Text-only) | 4.60 | 4.10 |
| N1 (Numerical-only) | 4.08 | 3.85 |
| T2 (Textualized Num + Text) | 3.66 | 3.42 |
| N2 (Num + Embedded Text) | 3.80 | 3.58 |
| *Training–objective / routing variants* | | |
| STL (Independent) | 4.28 | 3.95 |
| MTL (Shared Encoder) | 4.12 | 3.78 |
| Heteroscedastic MTL | 3.85 | 3.52 |
| Routing (Soft, Learned Paths) | 3.62 | 3.34 |

## 4.3 Generalization to other domains

To evaluate the generality of our routing framework, we applied it to three diverse datasets: Yelp Reviews (business feedback, 5,000 samples), RateMyProfessor (academic evaluations, 1,000 samples), and Customer Satisfaction (structured service surveys, 2,000 samples). All contain both structured numeric inputs (e.g., ratings, survey scores) and unstructured text (e.g., freeform comments), aligning with the multimodal multitask structure of our method. We compare three configurations: soft routing (full model), hard routing, and a non-routing ensemble.

Across all datasets, the full model with soft routing achieves the best performance. Hard routing performs slightly worse, with an average drop of 0.5–2.0 percentage points in $R^2$. Ablation studies further show that the text-based paths (T1 and T2) are the most critical, while N2 provides useful complementary information.

## 5 Discussion and Limitations

**Discussion.** Our experiments show that an adaptive routing framework can improve multimodal multitask prediction under substantial input and task heterogeneity. First, the synthetic benchmark confirms that the router discovers the latent mapping between modality relevance and task correlation. The model concentrates probability mass on paths that match the ground-truth data-generation rules and yields lower RMSE than static baselines. Second, on the psychotherapy data, the framework selects the textualized-numeric path (T2) for samples with rich notes and numeric scores and prefers the numeric-only path (N1) when notes provide little extra signal. Routing probabilities align with clinical expectations: patients whose therapist notes give detailed symptom descriptions rely more on

Table 3: Generalization across domains. The full model with soft routing achieves the best performance across datasets. Hard routing shows a small degradation (0.5–2.0 percentage points), while removing routing entirely causes a substantial drop ($\sim$20–25%).

| Dataset | Config | $R^2$ (Task 1) | $R^2$ (Task 2) | $\Delta$ vs. Full Model |
|---|---|---|---|---|
| **Yelp** | Soft Routing | 0.2406 | 0.2263 | — |
| | Hard Routing | 0.2350 | 0.2200 | $-0.0056$ ($-2.3\%$) |
| | No Routing | 0.1790 | 0.1651 | $-0.0616$ ($-25.6\%$) |
| **RateMyProf** | Soft Routing | 0.2850 | 0.2720 | — |
| | Hard Routing | 0.2800 | 0.2670 | $-0.0050$ ($-1.8\%$) |
| | No Routing | 0.2200 | 0.2100 | $-0.0650$ ($-22.8\%$) |
| **Customer Sat.** | Soft Routing | 0.3200 | 0.3050 | — |
| | Hard Routing | 0.3150 | 0.3000 | $-0.0050$ ($-1.6\%$) |
| | No Routing | 0.2500 | 0.2400 | $-0.0700$ ($-21.8\%$) |

language models, while patients with minimal notes rely on structured scores. Third, uncertainty-aware losses complement routing. The heteroscedastic objective discounts high-variance samples and reduces over-fitting, particularly for PHQ-9, which empirically exhibits greater label noise. Further, the mixture-of-experts architecture yields interpretable sub-networks. Each expert specializes in a clearly defined modality–task configuration, which supports post-hoc inspection and potential deployment in clinical workflows where transparency is essential.

**Limitations.**   The study relies on substantial synthetic data to stress-test routing behavior under controlled heterogeneity. Synthetic features follow idealized distributions, carry noise-free labels, and may embed patterns that rarely occur in real notes or questionnaire responses. Also, our "real-world" evaluation uses a hybrid corpus that augments genuine records with synthetically generated encounters, which expands data volume but creates distribution drift. Future work should validate on large untouched real-only test sets, apply importance weighting or domain-adversarial fine-tuning to reduce synthetic bias, and recalibrate class probabilities to reflect true clinical prevalence.

## Acknowledgments and Disclosure of Funding

This research is supported by funding from the National Institute of Aging (P30AG073105; a2Collective; Solomonov, Grosenick), the National Institute of Mental Health (R01MH131534; Grosenick)(K23 MH123864; Solomonov), NIH ORIP (R01OD039830; Grosenick), the Cornell Center for Pandemic Prevention and Response (CCPPR) (Grosenick, Solomonov), and the Brain and Behavior Research Foundation (NARSAD Young Investigator Grant; Solomonov).

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

# A  Real-world data collection and usage

**Realistic Synthetic Data.**  The clinical outcomes described here were not used directly for model training or evaluation in the main manuscript. Instead, they served as the basis of a highly realistic synthetic dataset that captures key distributional and structural characteristics of the real data.

**Ethical oversight.**  Weill Cornell Medicine's institutional review board approved the study; a waiver of informed consent was obtained for this retrospective study because it posed no more than minimal risk, did not affect care, rights or welfare and was deidentified for the purpose of analyses.

**Intervention and Data Collection**

- **Context.** A four-session, remote, skills-based psychotherapy program was launched at a large academic medical center during the first peak of the COVID-19 pandemic (March 2020–April 2021).

- **Participants.** $N = 534$ health care workers (HCWs) self-referred for emotional support. Role categories (percentages approximated): 35.2% nursing, 24.3% patient support, 22.8% administrative, 13.8% medical trainees/faculty, 2.4% facilities, 1.3% family members. 70% were on-site (frontline); 19% worked remotely; 11% unspecified.

- **Clinicians.** Sixty-seven trained providers (licensed psychologists, psychiatrists, social workers, and supervised trainees) delivered a total of 1,423 telehealth sessions.

- **Measures.**
    - Patient Health Questionnaire–9 (PHQ-9) and Generalized Anxiety Disorder–7 (GAD-7) at sessions 1 and 4.
    - PHQ-4 at sessions 2 and 3 for interim symptom tracking.
    - Columbia Suicide Severity Rating Scale (C-SSRS) at intake and as needed for suicide risk.

- **Safety.** Participants with severe symptoms or safety concerns were referred to appropriate emergency or long-term psychiatric care.

A linear mixed-effects model (random intercept and slope per subject; fixed effect of time) showed significant reduction in overall symptom burden:

$$\text{PHQ-4}_{\text{baseline}} = 5.65 \pm 2.95 \longrightarrow \text{PHQ-4}_{\text{final}} = 3.32 \pm 2.46,$$

$$F_{3,823} = 109.23, \ p < .001, \ \eta^2_{\text{partial}} = 0.27.$$

For participants with clinically elevated symptoms at baseline (PHQ-4 $\geq 6$), the effect size was stronger ($\eta^2_{\text{partial}} = 0.46$). Response rates were 42% on GAD-7 and 43% on PHQ-9 ($\geq 50\%$ symptom reduction).

# B  Computational Efficiency

Our model incorporates a hierarchical routing system to dynamically assign each sample to one of eight expert pathways, defined by the Cartesian product of four modality configurations (T1, T2, N1, N2) and two task strategies (STL, MTL). We analyze the computational complexity introduced by this routing mechanism relative to fixed single-path baselines.

Let $P$ denote the computational cost of a single expert model (i.e., the cost of a forward and backward pass through one pathway), and let $R$ denote the cost of computing the routing distributions using the modality and task routers. The routers are shallow MLPs whose cost is negligible relative to $P$, so $R \ll P$.

**Fixed Baseline (No Routing).** In the simplest configuration, a sample is passed through a fixed expert (e.g., modality T1 with STL). The total cost per sample is

$$C_{\text{fixed}} = \Theta(P).$$

**Soft Routing.** In soft routing, all eight experts are evaluated per sample. The model computes weighted predictions by combining each expert's output according to the product of modality and task routing probabilities. Thus, the total cost per sample is:

$$C_{\text{soft}} = R + \sum_{i=1}^{4} \sum_{j=1}^{2} P^{(i,j)}.$$

Assuming each expert has similar cost $P$, we have:

$$C_{\text{soft}} = R + 8P = \Theta(8P + R) \approx \Theta(8P).$$

**Hard Routing.** In hard routing, a sample is assigned to a single expert using a differentiable approximation of the argmax operation (e.g., via Gumbel-softmax). Only one expert is evaluated, resulting in:

$$C_{\text{hard}} = R + P = \Theta(P + R) \approx \Theta(P).$$

**Relative Overhead.** Compared to a fixed-path baseline, soft routing incurs an $\sim 8\times$ increase in per-sample computational cost, while hard routing maintains nearly equivalent cost with a marginal increase due to the routing networks. A summary is given in Table 4.

Table 4: **Per-sample computational cost of different routing strategies.**

| Routing Strategy | Experts Evaluated | Cost per Sample | Relative to Fixed |
|---|---|---|---|
| Fixed Path (e.g., T1+STL) | 1 | $\Theta(P)$ | 1 |
| Hard Routing | 1 | $\Theta(P + R)$ | $\approx 1\times$ |
| Soft Routing | 8 | $\Theta(8P + R)$ | $\approx 8\times$ |

In practice, training under soft routing enables all expert paths to specialize jointly and contributes to improved generalization. However, at inference time, switching to hard routing allows for efficient deployment by evaluating only a single expert path per input. This separation between training and inference modes enables a favorable trade-off between flexibility and computational efficiency.

## C    Scaling to More Tasks and Modalities

Our probabilistic routing architecture introduces non-trivial scaling behavior as the number of input modalities and prediction tasks increases.

Let $m$ be the number of input modalities. For each modality, we support bidirectional transformations to every other modality, enabling cross-modal fusion. This results in $m(m-1)$ transformation-based fusion paths. Including one native path per modality, the total number of modality-specific pathways becomes $m^2$.

Let $t$ be the number of outcome tasks. We consider three task-structuring strategies:

1. **STL + single MTL:** Each modality path is paired with $t$ STL heads and one MTL head covering all tasks. The total number of experts is
$$E = m^2 \times (t + 1).$$

2. **STL + pairwise MTL:** Instead of one MTL model, we train a separate MTL model for each task pair. The total number of experts becomes
$$E = m^2 \times \left( t + \frac{t(t-1)}{2} \right) = m^2 \times \frac{t(t+1)}{2},$$
which grows quadratically with the number of tasks.

3. **STL + grouped MTL (recommended for scalability):** Tasks are grouped into $g$ semantically related clusters, each modeled by a shared encoder with group-specific heads. The total number of experts becomes
$$E = m^2 \times (t + g), \quad \text{where } g \ll \frac{t(t-1)}{2}.$$
This strikes a balance between parameter efficiency and the ability to model inter-task relationships.

Let $P$ denote the computational cost of a single expert (forward/backward pass), and $R$ the cost of computing routing distributions. In soft routing, all experts are evaluated and weighted per sample. The total per-sample cost is

$$C_{\text{soft}} = \Theta(E \cdot P + R),$$

where $E$ is the total number of experts. For example:

- Under STL + single MTL: $C_{\text{soft}} = \Theta\big(m^2 \cdot (t+1) \cdot P\big)$

- Under STL + pairwise MTL: $C_{\text{soft}} = \Theta\Big(m^2 \cdot \frac{t(t+1)}{2} \cdot P\Big)$

In hard routing, only one expert is evaluated per sample (selected via argmax or Gumbel-softmax). The cost reduces to

$$C_{\text{hard}} = \Theta(P + R),$$

independent of the total number of experts, though all expert parameters must still reside in memory.

## D  Data-Driven (Empirical) Synthetic Data

To support controlled experimentation and model probing, we generated synthetic samples that reflect "clean" examples from the original dataset. These samples were designed to preserve strong predictive signal across both GAD and PHQ outcomes. The process is structured to maintain cross-modal coherence between structured numerical features and free-text clinical notes. The generation process proceeds in three stages: (1) Sampling structured numerical features from a smoothed approximation of high-confidence empirical distributions. (2) Generating text conditionally based on these numerical features using a large language model (LLM) with custom prompting. (3) Filtering generated samples using our trained multimodal multitask model to retain only those with low predictive uncertainty and high task-specific confidence.

### D.1  Synthetic Numerical Data Generation

We generated synthetic numerical data using three methods, each designed to preserve the statistical structure of the original dataset.

**Gaussian Synthesis.** We estimated the mean vector $\boldsymbol{\mu} \in \mathbb{R}^d$ and covariance matrix $\boldsymbol{\Sigma} \in \mathbb{R}^{d \times d}$ from the original dataset. To ensure numerical stability, a small constant $10^{-6}$ was added to the diagonal of $\boldsymbol{\Sigma}$. Synthetic samples were then drawn from a multivariate normal distribution:

$$x_{\text{synthetic}} \sim \mathcal{N}(\boldsymbol{\mu}, \boldsymbol{\Sigma}), \quad x_{\text{synthetic}} \in \mathbb{R}^{n_{\text{samples}} \times d}.$$

**Copula-Based Synthesis.** We applied a rank-based transformation to map each feature to a standard normal distribution. The empirical correlation matrix was computed on the transformed data. Synthetic samples were drawn from a multivariate normal distribution with this correlation structure and subsequently mapped back to the original marginal distributions using a quantile transform. This method preserved nonlinear dependencies among features.

**Kernel Density Estimation (KDE) Synthesis.** Continuous features were standardized using `StandardScaler`. A Gaussian kernel density estimator was fitted with bandwidth $h = n^{-1/(d+4)}$, where $n$ is the number of samples and $d$ is the feature dimensionality. New samples were drawn using KDE-based resampling. Binary features (e.g., *dissociate*, *anger*, *fear_contam*) were thresholded post-generation:

$$x_{\text{synthetic}}[\text{binary}] = (x_{\text{synthetic}}[\text{binary}] > 0.5)\texttt{.astype(int)}.$$

For each method, we generated 200 synthetic samples while maintaining the original class distribution of PHQ-9 and GAD-7 binary outcomes. The synthetic datasets closely matched the original statistical characteristics, with a mean absolute difference in feature correlation of less than 0.1 and a KL divergence in class distribution of less than 0.05.

**Synthetic Text Data Generation**

To generate synthetic text data that reflect realistic psychotherapy session notes, we employed a prompt-based generation pipeline using small, open-source large language models (LLMs). The goal was to create natural language samples that are consistent with the patterns observed in the original dataset, while maintaining data privacy and avoiding memorization of sensitive content.

**Numerical-to-Text Conversion.** We first transformed structured numerical features into short natural language descriptions. For each synthetic numerical instance $x \in \mathbb{R}^d$, we constructed a template-based summary containing key symptom indicators and severity scores (e.g., PHQ-9, GAD-7). An example of this intermediate representation is:

> *The patient reported a PHQ-9 score of 15 and a GAD-7 score of 13. They endorsed symptoms such as dissociation and irritability, with no signs of fear of contamination.*

**LLM-Based Natural Language Expansion.** We used an instruction-tuned language model (Flan-T5 or Phi-2) to expand the structured summaries into fluent and contextually appropriate psychotherapy notes. Each prompt followed the format:

> *Patient data: PHQ-9 = 15, GAD-7 = 13, symptoms = [dissociation, irritability]. Write a brief therapist note summarizing the patient's emotional state and challenges.*

The model generated coherent text such as:

> *The patient presented with moderate symptoms of depression and anxiety, including dissociative experiences and heightened irritability. They expressed difficulty managing emotional stressors and reported low energy and trouble sleeping.*

**Post-Processing and Filtering.** We generated one therapist-style note per synthetic numerical input, resulting in 200 synthetic text samples. To ensure linguistic diversity and clinical plausibility, we applied basic heuristics to filter out degenerate outputs (e.g., overly repetitive or off-topic content). The vocabulary and sentence structure were qualitatively consistent with those in the original data, and generated texts maintained semantic alignment with the associated synthetic numerical features.

**Privacy Considerations.** All models were run locally without external API calls to ensure HIPAA compliance. We used only open-access, instruction-tuned models with small memory footprints (Phi-2), which allowed controlled offline generation and ensured that no real patient data were exposed or used during synthesis.

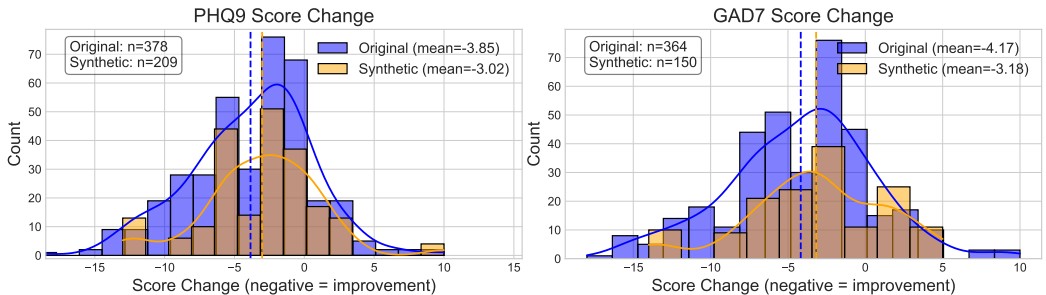

Figure 5: Distribution of synthetic and original data

# E    Equation-Driven (Analytical) Synthetic Data

Our synthetic benchmark is designed to rigorously test input-dependent routing in multitask, multimodal regression. Each sample consists of two modalities, a numeric vector and a text vector, and two regression targets. Both modalities contribute to both tasks, but the degree and nature of their informativeness is heterogeneous and input-dependent, simulating real-world complexity.

**Input Features.** For each sample, we generate:

- Numeric features $x^{(\text{num})} \in \mathbb{R}^{d_{\text{num}}}$, sampled as $\mathcal{N}(0, I)$.
- Text features $x^{(\text{text})} \in \mathbb{R}^{d_{\text{text}}}$, also sampled as $\mathcal{N}(0, I)$.

We use $d_{\text{num}} = d_{\text{text}} = 16$ unless otherwise specified.

**Random Fourier Feature Maps.** To introduce nonlinear, cross-modal dependencies, we use random Fourier feature (RFF) maps:

$$\phi(x^{(\text{text})}) = \sqrt{\frac{2}{D}} \cos(W_\phi x^{(\text{text})} + b_\phi),$$

$$\psi(x^{(\text{num})}) = \sqrt{\frac{2}{D}} \cos(W_\psi x^{(\text{num})} + b_\psi),$$

where $W_\phi, W_\psi \in \mathbb{R}^{D \times 16}$ have entries drawn from $\mathcal{N}(0, 1)$, $b_\phi, b_\psi \in \mathbb{R}^D$ are drawn uniformly from $[0, 2\pi]$, and $D = 32$.

**Target Construction.** For each sample, we generate two targets:

$$y_1 = \alpha_1^\top x^{(\text{num})} + \beta_1^\top \phi(x^{(\text{text})}) + \gamma_1 \cdot \sin(\omega_1^\top x^{(\text{num})}) + \epsilon_1,$$
$$y_2 = \alpha_2^\top x^{(\text{text})} + \beta_2^\top \psi(x^{(\text{num})}) + \gamma_2 \cdot \cos(\omega_2^\top x^{(\text{text})}) + \epsilon_2,$$

where:

- $\alpha_k, \beta_k \sim \mathcal{N}(0, I)$ (dimensions match their arguments)
- $\omega_k \sim \mathcal{N}(0, I)$ (dimension 16)
- $\gamma_k \sim \text{Uniform}[0.5, 1.5]$
- $\epsilon_k \sim \mathcal{N}(0, 0.1^2)$.

All parameters are independently sampled for each trial, and fixed for all samples within a trial.

**Design Rationale.** This construction ensures:

- Both modalities are relevant to both tasks, but in different, nonlinear, and cross-modal ways.
- The use of RFFs simulates learned embeddings and increases the complexity of the mapping.
- Sinusoidal nonlinearities further challenge the model, requiring it to capture nontrivial dependencies.
- The setup allows for controlled ablations (e.g., by zeroing coefficients or removing nonlinear terms).

**Implementation Notes.**

- All random seeds are fixed for reproducibility.
- The code for data generation is provided in the supplementary repository.
- The synthetic dataset can be easily extended to more modalities or tasks by following the same recipe.

---

**Algorithm 1** Synthetic Data Generation

---

1: Sample $x^{(\text{num})}, x^{(\text{text})} \sim \mathcal{N}(0, I_{16})$
2: Compute $\phi(x^{(\text{text})}), \psi(x^{(\text{num})})$ via RFFs
3: Sample $\alpha_k, \beta_k, \omega_k, \gamma_k$ as above
4: Compute $y_1, y_2$ as above, add noise $\epsilon_k$

---

```
def rff(x, W, b):
return np.sqrt(2 / W.shape[0]) * np.cos(W @ x + b)

x_num = np.random.randn(16)
x_text = np.random.randn(16)
W_phi, b_phi = np.random.randn(32, 16), np.random.uniform(0, 2*np.pi, 32)
W_psi, b_psi = np.random.randn(32, 16), np.random.uniform(0, 2*np.pi, 32)
phi_x_text = rff(x_text, W_phi, b_phi)
psi_x_num = rff(x_num, W_psi, b_psi)
# ... sample coefficients and compute y1, y2 as above
```

## E.1   Scenario 1: Sinusoidal/Cosine, Both Modalities

```
Number of samples: 1000 (train), 1000 (test)
Feature dimensions: d_num = 16, d_text = 16, D (RFF output) = 32

Feature maps:
phi(x_text) = sqrt(2/32) * cos(W_phi x_text + b_phi)
psi(x_num) = sqrt(2/32) * cos(W_psi x_num + b_psi)
W_phi, W_psi ~ N(0, 1), b_phi, b_psi ~ Uniform[0, 2pi]

Target equations:
y1 = alpha1^T x_num + beta1^T phi(x_text) + gamma1 * sin(omega1^T x_num) + epsilon1
y2 = alpha2^T x_text + beta2^T psi(x_num) + gamma2 * cos(omega2^T x_text) + epsilon2

Parameter distributions:
 - alpha_k, beta_k ~ N(0, I)
 - omega_k ~ N(0, I)
 - gamma_k ~ Uniform[0.5, 1.5]
 - epsilon_k ~ N(0, 0.1^2)
```

In this scenario, as expected, the model learns that using MTL yields the best performance (see Figure 6).

## E.2   Scenario 2: STL Preferred

```
Number of samples: 1000 (train), 1000 (test)
Feature dimensions: d_num = 16, d_text = 16, D (RFF output) = 32

Feature maps:
phi(x_text) = sqrt(2/32) * cos(W_phi x_text + b_phi)
psi(x_num) = sqrt(2/32) * cos(W_psi x_num + b_psi)
W_phi, W_psi ~ N(0, 1), b_phi, b_psi ~ Uniform[0, 2pi]

Target equations:
y1 = alpha1^T x_num + gamma1 * sin(omega1^T x_num) + epsilon1
y2 = alpha2^T x_text + gamma2 * cos(omega2^T x_text) + epsilon2

Parameter distributions:
 - alpha_k ~ N(0, I)
 - omega_k ~ N(0, I)
 - gamma_k ~ Uniform[0.5, 1.5]
 - epsilon_k ~ N(0, 0.1^2)
```

In this scenario, each task is generated from a single modality: $y_1$ depends only on the numeric features and their nonlinear transformation, while $y_2$ depends only on the textual features and their nonlinear transformation. Since each task is generated independently from its own modality, there is no shared information or benefit to learning the tasks jointly. As a result, the model learns that treating each task separately by using STL yields the best performance (see Figure 7).

### E.3 Scenario 3: Fusion-Dominant Routing

```
Number of samples: 1000 (train), 1000 (test)
Feature dimensions: d_num = 16, d_text = 16, D (RFF output) = 32

Feature maps:
phi(x_text) = sqrt(2/32) * cos(W_phi x_text + b_phi)
psi(x_num) = sqrt(2/32) * cos(W_psi x_num + b_psi)
W_phi, W_psi ~ N(0, 1), b_phi, b_psi ~ Uniform[0, 2pi]

Target equations:
y1 = alpha1^T x_num + beta1^T phi(x_text)  + epsilon1
y2 = alpha2^T x_text + beta2^T psi(x_num)  + epsilon2

Parameter distributions:
 - alpha_k, beta_k ~ N(0, I)
 - omega_k ~ N(0, I)
 - epsilon_k ~ N(0, 0.1^2)
```

The results in Figure 8show that the model prefers the T2 (fusion) modality path, especially in combination with the MTL (multi-task learning) paradigm. The N1 and N2 paths are rarely or never used for MTL, indicating that the model has learned to avoid these routes in favor of more effective ones.

## F   Broader Impact

This work introduces a flexible machine learning framework for adaptively routing data through multimodal and multitask pathways, with primary application to psychological outcome prediction. By enabling models to select personalized computation paths based on both input availability and task structure, this approach has the potential to improve prediction accuracy and robustness in real-world settings where data is heterogeneous and incomplete.

While our evaluation is framed in the context of mental health prediction, the methodology is broadly applicable to domains such as clinical decision support, education, and human-centered AI systems where structured and unstructured inputs coexist and multiple outcomes must be considered jointly.

At the same time, predictive models in healthcare and mental health raise significant ethical concerns. These include the risk of reinforcing biases present in clinical documentation, the opacity of model decisions, and the potential for overreliance on algorithmic outputs in high-stakes scenarios. Our model attempts to mitigate some of these risks by producing interpretable routing decisions, which may offer insight into modality usefulness and task interactions. Nonetheless, interpretability and fairness should be further studied before deployment.

Our work uses synthetic data and carefully preprocessed clinical data to demonstrate technical contributions, and does not aim to inform clinical decisions directly. Future use of this method in real-world applications must be coupled with appropriate clinical validation, governance, and safeguards to ensure equitable, transparent, and accountable outcomes.

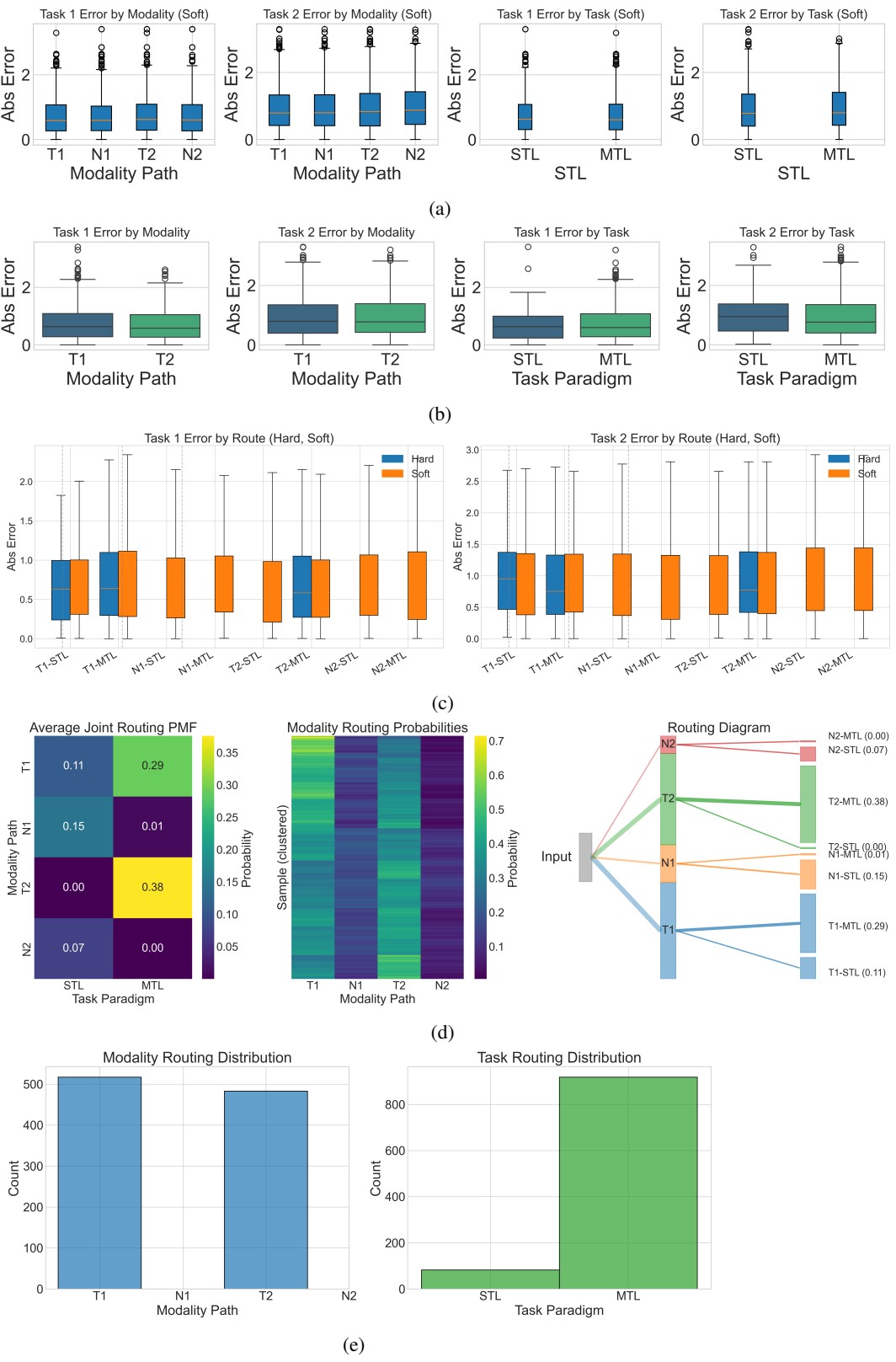

Figure 6: **Scenario 1: Sinusoidal/Cosine, Both Modalities** (a) Absolute error by route (soft routing, weighted by probabilities). (b) Absolute error by route (hard routing, based on most probable path). (c) Comparison of absolute errors for each route: hard vs. soft routing (note that in hard routing, boxes do not appear in every case as the router can learn to bypass N1 as a less probable path). (d) Summary: joint routing PMF, clustered heatmap, and routing Sankey diagram. (e) Distribution of selected modality and task paradigm by the router.

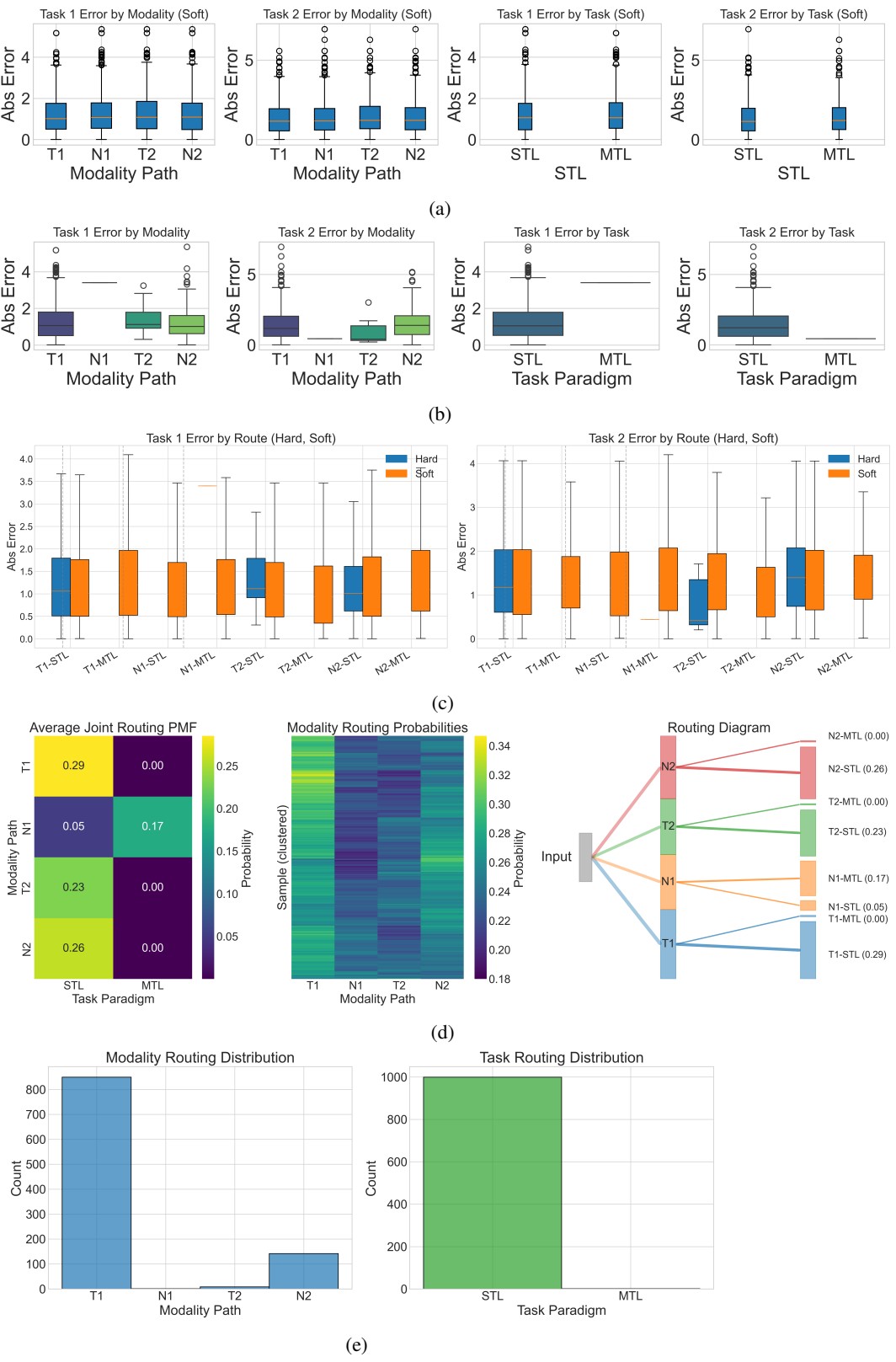

Figure 7: **Scenario 2: STL Prefered** (a) Absolute error by route (soft routing, weighted by probabilities). (b) Absolute error by route (hard routing, based on most probable path). (c) Comparison of absolute errors for each route: hard vs. soft routing (note that in hard routing, boxes do not appear in every case). (d) Summary: joint routing PMF, clustered heatmap, and routing Sankey diagram. (e) Distribution of selected modality and task paradigm by the router.

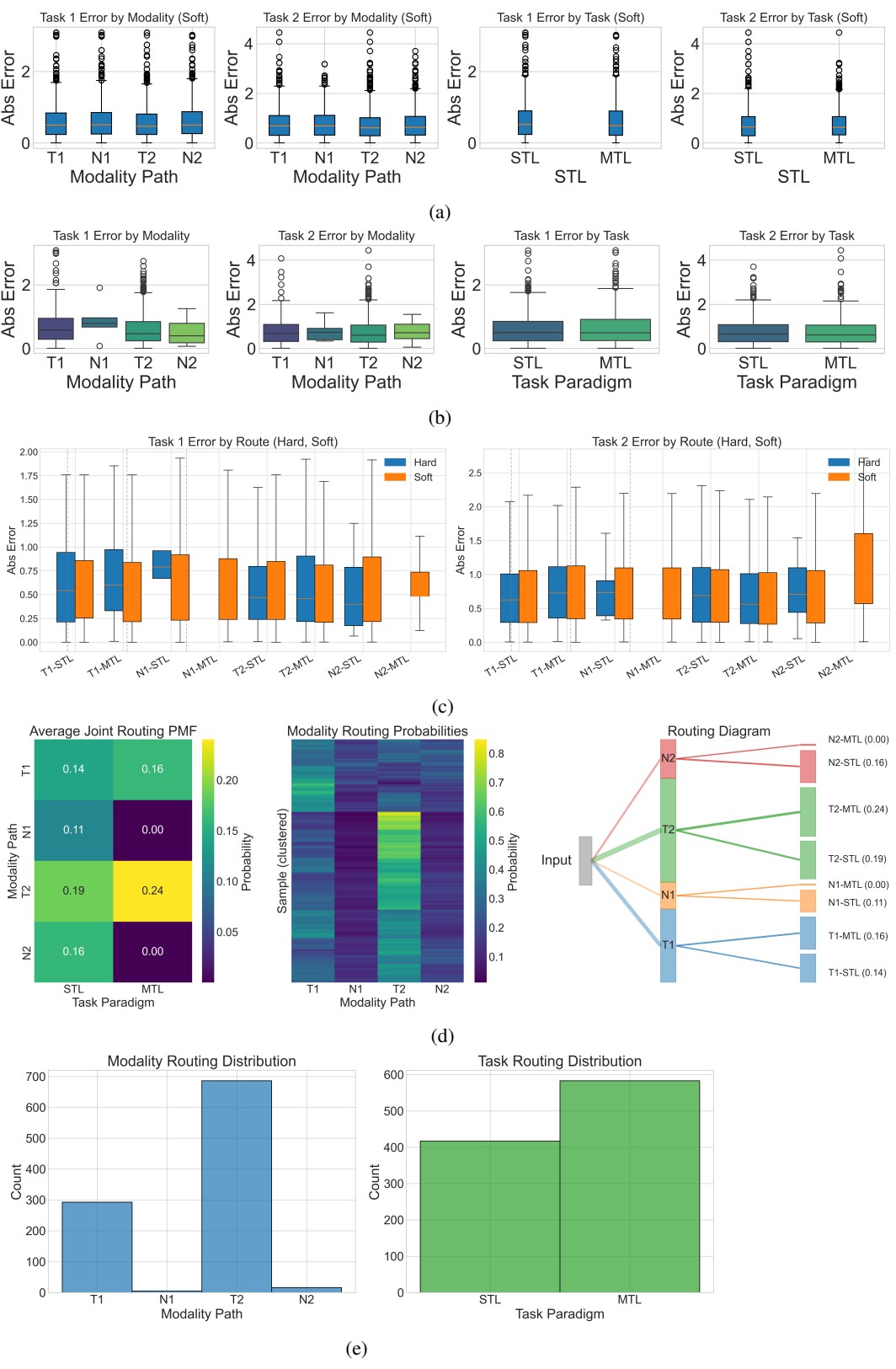

Figure 8: **Scenario 3: Fusion-Dominant Routing** (a) Absolute error by route (soft routing, weighted by probabilities). (b) Absolute error by route (hard routing, based on most probable path). (c) Comparison of absolute errors for each route: hard vs. soft routing (note that in hard routing, boxes do not appear in every case). (d) Summary: joint routing PMF, clustered heatmap, and routing Sankey diagram. (e) Distribution of selected modality and task paradigm by the router.

