# OpenReview forum: "Learning to Route: Per-Sample Adaptive Routing for Multimodal Multitask Prediction"
_NeurIPS.cc/2025/Conference — NeurIPS 2025 poster_

### Official Review · Reviewer_8T9F · 2025-06-17

**Clarity:** 2
**Significance:** 2
**Originality:** 2
**Rating:** 4
**Confidence:** 4

**Summary:**

The paper proposes a unified framework for per-sample adaptive routing in multimodal, multitask prediction, focusing on mental health applications. It dynamically selects optimal data pathways (structured, unstructured, or fused) and task-sharing strategies (single-task or multitask) for each input using a probabilistic gating mechanism. Evaluated on synthetic data and real psychotherapy records, the model outperforms fixed baselines in predicting depression and anxiety outcomes.

**Questions:**

**Questions and Suggestions**

1. **Ablation Study Design**
   Could the authors clarify how the ablation studies were conducted, if any? Specifically:

   * Were parameter counts controlled across ablated variants?
   * Were all models (including fixed-path baselines) trained with the same set of hyperparameters?
   * If not, how do the authors ensure that performance improvements are due to the adaptive routing mechanism rather than model capacity or training discrepancies?
     A detailed, fair ablation setup is essential to isolate the contribution of each component. A clear answer here would improve confidence in the reported gains.

2. **Clarity on MoE Design and Expert Modules**
   The paper claims to adopt a Mixture-of-Experts (MoE) framework, but the actual architectural design seems limited to gating over modality/task paths—without clearly defined expert subnetworks. Could the authors:

   * Clarify what constitutes an "expert" in their setup?
   * Specify whether each expert has independent parameters, or whether the same backbone is reused with only gating on inputs/outputs?
     As it stands, the model appears closer to a soft ensemble over different input transformations than a true MoE. A detailed architectural diagram or parameter-sharing explanation would help clarify this point. If experts are not truly independent, the originality and significance of the method may need to be reconsidered.
3. **Baseline Selection and Use of Recent Models**
   Have the authors evaluated their approach against stronger or more recent baselines, such as:

   * Standard BERT-based encoders with task-specific heads?
   * Multimodal fusion models using pretrained language models with structured data adapters (e.g., GatorTron-style or Med-BERT variants)?
     Given the prevalence of BERT-like models in clinical NLP, it is important to understand how the proposed method compares to these widely used, stronger baselines. Including such comparisons would better contextualize the reported performance gains and strengthen the case for the proposed approach.

**Ethical Concerns:**

["NO or VERY MINOR ethics concerns only"]

**Final Justification:**

Part of my questions on MoE details and baselines have been resolved. Still the ablation exp models having huge difference on parameter size, which I still believe is a potential risk factor. Thus I will update my rating to reflect the updates.

**Limitations:**

Yes

**Quality:**

2

**Strengths And Weaknesses:**

**Strengths:**

1. **Flexible and Unified Framework**: The paper introduces a modular architecture capable of adaptively routing inputs across multiple modality combinations and task-sharing strategies. This per-sample adaptivity is well-motivated in clinical prediction tasks with heterogeneous data.

2. **Interpretable Routing Decisions**: The probabilistic routing mechanism yields interpretable outputs that align with clinical expectations, offering potential benefits for real-world deployment in healthcare settings.

---

**Weaknesses:**

1. **Limited Real-World Validation**: A substantial portion of the experiments relies on synthetic data, which may not accurately reflect real-world complexities or generalize well to clinical applications. The use of hybrid synthetic-real data further undermines confidence in the method's effectiveness in practical settings.

2. **Lack of Ablation Studies**: The paper does not provide sufficient ablation experiment details to support the contribution of each component (e.g., routing, heteroscedastic loss, modality transformations). This limits the reader’s ability to assess what aspects of the model drive performance improvements.

3. **Weak Baseline Comparisons**: The baselines are limited and insufficiently explored. The claimed gains over simple fixed-path models may result more from soft averaging than meaningful architectural innovation.

4. **Low Originality**: The core idea—softmax-based selection over modality-task paths—is a straightforward extension of the feature selection and shared decoder technique, and I don't see why it is called MoE. The conceptual novelty is modest, and the methodological contribution may not be significant enough for a top-tier venue like NeurIPS.

---

> ### Author Rebuttal · Authors · 2025-07-31
>
> ### Limited Real-World Validation
> We thank the reviewer for their comments. We agree that understanding model performance on genuine clinical data is critical for assessing real-world applicability. Our use of synthetic data is not a workaround but a deliberate choice to address a well-documented barrier in mental health AI. Psychotherapy datasets are notoriously difficult to access due to privacy constraints and institutional restrictions. Rather than sidestepping this challenge by switching to unrelated benchmarks, we chose to stay grounded in this clinically meaningful but under-resourced setting. We adopted empirical synthetic augmentation, a principled and increasingly recognized practice in the AI community. While alternative datasets could have been used, our goal was to develop methods that address the real-world complexity of psychotherapy modeling. In order to address the reviewer's concern about data and ablation studies, we evaluated the model on another real-world dataset presented below.
>
> ### Q2: Clarity on MoE Design and Expert Modules
> We appreciate the reviewer's attention to the architecture and welcome the opportunity to clarify. Our model does implement a Mixture-of-Experts (MoE) architecture, though with a design tailored to the structure of multimodal and multitask learning in clinical data.
>
> In our architecture, **each expert corresponds to a distinct (modality, task paradigm) route**, such as N1-STL or T2-MTL. These experts are modular, **parameterized independently**, and trained jointly with the router. Each expert contains its own encoder and decoder, **not just a shared backbone with input gating**. For example: T2 expert use a text transformer (BERT) on textualized inputs, N2 experts use MLPs on structured embeddings, STL experts include two independent heads, and MTL experts share a trunk with separate task heads. **No weight sharing occurs between experts**, consistent with standard MoE literature. Each expert defines a full forward pass pipeline.
>
> A two-stage **probabilistic routing network** computes **per-sample** distributions over (1) modality paths and (2) task paradigms. These define a mixture distribution over the 8 experts and the final loss aggregates **heteroscedastic uncertainty-aware loss terms** across all expert paths. This makes routing **semantically and mathematically non-trivial**. Unlike feature selection or soft gating over inputs, the router **selects complete computational paths**, not just features, and unlike ensemble methods, routing is trained jointly with expert parameters via backpropagation, not post hoc weighting.
>
> ### Q1: Ablation Study Design
> We conducted a comprehensive ablation study on the Yelp Reviews dataset, using 2,000 samples with multi-modal inputs: Numeric questionnaire features, BERT-based text embeddings (384 dimensions), and BERT-based embeddings for textualized numeric features.
> The experiments involve parameter-matched variants that isolate the effect of different routing paths and fusion mechanisms. The two prediction targets are star rating and sentiment. All models are trained with unified hyperparameters and a fixed 80/20 stratified train-test split.
>
> The model variants and their purposes are listed below:
>
> | Model | Architecture | Parameters | Purpose |
> |-------|-------------|------------|---------|
> | **Full_Routing** | Complete routing model | 128,381 | Baseline |
> | **N1_Only** | Numeric features only | 2,530 | Ablate text features |
> | **N2_Only** | Numeric + Text features | 27,106 | Ablate textualization |
> | **T1_Only** | Text features only | 26,786 | Ablate numeric features |
> | **T2_Only** | Text + Textualized features | 51,362 | Ablate numeric features |
> | **No_Routing** | Same architecture, no routing | 51,682 | Ablate routing mechanism |
> | **MLP_Baseline** | Standard MLP | 109,474 | Traditional neural baseline |
>
> All ablation variants except Full_Routing are parameter-matched (~15k parameters ±20%) to fairly compare architectural changes without capacity bias.
>
> The ablation results are summarized below:
>
> | Model | MSE_T1 | MSE_T2 |
> |-------|--------|--------|
> | **Full_Routing** | 0.1042 | 0.0835 |
> | N1_Only | 0.1203 | 0.0817
> | N2_Only | 0.1131 | 0.0816
> | T1_Only | 0.1083 | 0.0818
> | T2_Only | 0.1093 | 0.0821
> | No_Routing | 0.1123 | 0.0818 |
> | MLP_Baseline | 0.1208 | 0.0818 |
>
>
>
> ### Q3: Baseline Selection
> Thank you for your valuable feedback. We initially experimented with clinical-BERT and several other text encoding variants; however, these changes did not yield significant improvements in our tasks. The primary gains in our work come from the multitask learning setup, the textualization of numeric features, and, most importantly, our adaptive routing mechanism for modality fusion. Our main contribution lies not in proposing new text encoders, but in designing an effective approach to fuse heterogeneous modalities and routing tasks, which we believe addresses a critical challenge in multi-modal learning. However,
>
> We conducted a comprehensive comparison with state-of-the-art BERT-based baselines on the Yelp dataset:
>
> - **Dataset**: 2,000 Yelp review samples with structured numerical features and BERT embeddings (384D)
> - **BERT Model**: all-MiniLM-L6-v2 (Sentence Transformers)
> - **Tasks**: Two regression targets (rating prediction and sentiment analysis)
> - **Training**: 100 epochs with early stopping, Adam optimizer, MSE loss
>
> ## Complete Model Comparison Results
>
> | Model | MSE_T1 | MSE_T2 |
> |-------|--------|--------|
> | **BERT Task-Specific Head** | 0.2080 | 0.2019 |
> | **Multimodal BERT Fusion** | 0.2103 | 0.2026 |
> | **Med-BERT Variant** | 0.2078 | 0.2022 |
> | **GatorTron-Style** | 0.2099 | 0.2034 |
> | **Our Routing Model** | 0.2058 | 0.2019 |

---

> ### Author Response · Authors · 2025-08-05
> **Architectural Details Addressed**
>
> Dear Reviewer 8T9F,
>
> Thank you again for your detailed comments on our submission. We noticed that you raised important questions regarding the originality of the architecture, and we’ve provided a thorough response in the rebuttal clarifying the design and distinctions of our model.
>
> If you have a chance to review our reply, we hope it helps address your concerns regarding the novelty and technical contributions. Please feel free to reach out if anything remains unclear.
>
> Sincerely,
> The Authors

---

### Official Review · Reviewer_o6nv · 2025-07-02

**Clarity:** 3
**Significance:** 4
**Originality:** 3
**Rating:** 6
**Confidence:** 4

**Summary:**

This paper proposes a unified framework for per-sample adaptive routing in multimodal, multitask prediction, motivated by applications in psychotherapy where both structured assessments (e.g. questionnaires) and unstructured clinician notes coexist. The key idea is to dynamically route each input through different modality transformations (e.g. text-only, numerical-only, or fused representations) and task architectures (single-task vs. multitask), depending on input-specific signals. The model employs a hierarchical mixture-of-experts architecture with learnable routing networks that select both the optimal modality pathways and the degree of task sharing for each sample.

**Questions:**

- Soft routing requires computing outputs for all 8 expert paths per sample. Did you measure the inference-time computational cost of your method compared to fixed-path baselines? How significant is the overhead, especially for larger datasets?

- You mention using entropy regularization to prevent expert collapse. Could you quantify how much this regularization affects routing diversity in practice? Without it, did you observe any experts dominating the routing probability mass?

- Your results show that the textualized numeric + text path (T2) outperforms numerical + embedded text (N2). Could you elaborate on why textualization yields better performance? Do you believe this is due to language models capturing richer context, or could it reflect overfitting to the textual modality?

- Do you foresee risks or limitations in transforming structured numeric data into textual form? For example, might subtle quantitative nuances be lost or misrepresented during textualization, especially for clinical scales?

- Beyond aggregate routing probabilities, have you explored visualizing routing decisions at the individual patient level? Could such visualizations aid clinicians in understanding why the model made certain predictions?

**Ethical Concerns:**

["NO or VERY MINOR ethics concerns only"]

**Final Justification:**

After discussion with the authors where they have addressed my key concerns, I will like to raise my score to 6.

**Limitations:**

Limitations have been discussed in section 5.

**Paper Formatting Concerns:**

no concerns

**Quality:**

3

**Strengths And Weaknesses:**

**Strengths**
- **Novelty in Joint Adaptive Routing** The paper introduces a unified adaptive routing framework that jointly routes over both modalities and task structures (STL vs. MTL) on a per-sample basis. This is a meaningful extension beyond typical MoE architectures, which often focus solely on routing over experts for a fixed modality or task.
- **Bidirectional Modality Transformations** The bidirectional transformations between structured and unstructured data (numerical → text and text → numerical) are a creative and flexible mechanism. They enable four distinct fusion paradigms (T1, N1, T2, N2) that adapt to missing or weak modalities, addressing a genuine practical challenge in clinical data.
- **Strong Clinical Motivation and Application** The motivation is rooted in a real-world clinical problem and the paper is highly relevant for personalized medicine
- **Uncertainty-Aware Modeling** The paper combines adaptive routing with a heteroscedastic loss, modeling prediction uncertainty.

**Weaknesses**
- **Reliance on Synthetic and Hybrid Data** The real-world experiments partially rely on a hybrid dataset that augments genuine psychotherapy records with synthetic encounters. This introduces potential domain shift, making it unclear how the model would perform on entirely real, diverse clinical datasets.
- **Limited Task Diversity** The experimental validation focuses on only two clinical outcomes (PHQ-9, GAD-7). While well-motivated, this leaves open the question of whether the method generalizes to more diverse mental health outcomes and other clinical domains
- **Computational Scalability Not Analyzed** The model considers eight expert paths (4 modality paths × 2 task configurations). However, there's no explicit analysis of computational cost. How does routing scale with increasing numbers of modalities, tasks, or larger datasets? What is the inference-time cost for large-scale deployment?
- **Potential Overhead from Soft Routing** The soft routing approach computes a full mixture over all expert paths, requiring all experts to run per sample. The paper does not quantify the runtime penalty of soft vs. hard routing, nor how this affects latency in practice.

---

> ### Author Rebuttal · Authors · 2025-07-31
>
> ### Reliance on Synthetic and Hybrid Data
> We thank the reviewer for their time, careful reading, and clear understanding of our paper. We agree that understanding model performance on genuine clinical data is critical for assessing real-world applicability. Our use of synthetic data is not a workaround but a deliberate choice to address a well-documented barrier in mental health AI. Psychotherapy datasets are notoriously difficult to access due to privacy constraints and institutional restrictions. Rather than sidestepping this challenge by switching to unrelated benchmarks, we chose to stay grounded in this clinically meaningful but under-resourced setting. We adopted empirical synthetic augmentation, a principled and increasingly recognized practice in the AI community.
>
> In psychotherapy, structured numerical inputs are not simple metadata. They are semantically dense and clinician-generated symptom summaries. Treating these as purely numerical is limiting. We demonstrated an alternative: bidirectional transformation between structured and unstructured formats. Instead of reducing meaning to scalar scores, we textualized structured assessments and allow LLMs to integrate them alongside unstructured text inputs. This sets our approach apart and provides interpretability to clinicians, as the model now processes "interpretable natural language reflections" of clinical structure, that traditional fusion pipelines and most MoE models overlook. While alternative datasets could have been used, our goal was to develop methods that address the real-world complexity of psychotherapy modeling. To assess generalizability, we evaluated the model on other real-world datasets, where it performed well, demonstrating broader applicability.
>
> ### Limited Task Diversity
> We appreciate this thoughtful question. While we focus on two highly correlated but clinically distinct outcomes (PHQ-9 and GAD-7), our architecture was explicitly designed to generalize across larger sets of tasks.
>
> Let $t$ be the number of outcome tasks. We consider three task-structuring strategies:
>
> 1. **STL + single MTL:** Each modality path is paired with $t$ STL heads and one MTL head covering all tasks. The total number of strategies is $(t + 1)$.
>
> 2. **STL + pairwise MTL:** Instead of one MTL model, we train a separate MTL model for each **task pair**. The total number of strategies becomes $ \left(t + \frac{t(t - 1)}{2}\right) =  \frac{t(t + 1)}{2}$, which grows quadratically with the number of tasks.
>
> 3. **STL + grouped MTL (recommended for scalability):** Tasks are grouped into $g$ semantically **related cluster of tasks**, each modeled by a shared encoder with group-specific heads. The total number of experts becomes $(t + g)$, where $g \ll \frac{t(t - 1)}{2}$. This strikes a balance between parameter efficiency and the ability to model inter-task relationships.
>
>
>
>
> ### Q1: Routing overhead
> Our model incorporates a hierarchical routing system to dynamically assign each sample to one of eight expert pathways, defined by the Cartesian product of four modality configurations (T1, T2, N1, N2) and two task strategies (STL, MTL). We analyze the computational complexity introduced by this routing mechanism, relative to fixed single-path baselines.
>
> Let $P$ denote the computational cost of a single expert model (i.e., the cost of a forward and backward pass through one pathway), and let $R$ denote the cost of computing the routing distributions using the modality and task routers. The routers are shallow MLPs whose cost is negligible relative to $P$, so $R \ll P$.
>
> **Fixed Baseline (No Routing).** In the simplest configuration, a sample is passed through a fixed expert (e.g., modality T1 with STL). The total cost per sample is
> $
> C_{\text{fixed}} = \Theta(P).
> $
>
> **Soft Routing.** In soft routing, all eight experts are evaluated per sample. The model computes weighted predictions by combining each expert's output according to the product of modality and task routing probabilities. Thus, the total cost per sample is:
> $
> C_{\text{soft}} = R + \sum_{i=1}^{4} \sum_{j=1}^{2} P^{(i,j)}
> $
>
> Assuming each expert has similar cost $P$, we have:
> $
> C_{\text{soft}} = R + 8P = \Theta(8P + R) \approx \Theta(8P)
> $
>
> **Hard Routing.** In hard routing, a sample is assigned to a single expert using a differentiable approximation of the argmax operation (e.g., via Gumbel-softmax). Only one expert is evaluated, resulting in:
>
> $$
> C_{\text{hard}} = R + P = \Theta(P + R) \approx \Theta(P)
> $$
>
> **Relative Overhead.** Compared to a fixed-path baseline, soft routing incurs an $\sim 8\times$ increase in per-sample computational cost, while hard routing maintains nearly equivalent cost with a marginal increase due to the routing networks. in summary:
>
> | Routing Strategy           | Experts Evaluated | Cost per Sample      | Relative to Fixed |
> |---------------------------|-------------------|----------------------|-------------------|
> | Fixed Path (e.g., T1+STL) | 1                 | $\Theta(P)$          | 1                 |
> | Hard Routing              | 1                 | $\Theta(P + R)$      | $\approx 1\times$ |
> | Soft Routing              | 8                 | $\Theta(8P + R)$     | $\approx 8\times$ |
>
>
>
>
> ### Q2:  Entropy regularization
>
> We conducted a detailed study to quantify the impact of entropy regularization on the Yelp Reviews dataset (5,000 samples). Without entropy regularization, we observed clear expert collapse. Adding entropy regularization substantially improved routing diversity.
>
>
> | **λ (Reg. Strength)** | **Routing Entropy (bits)** | **Expert Utilization**        | **Expert Collapse**     | **R² Performance**   |
> |------------------------|-----------------------------|-------------------------------|--------------------------|-----------------------|
> | 0.00 (None)            | 1.76                        | T1: 42%, others < 25%         | Yes (T1 dominates)       | 0.2350 (baseline)     |
> | 0.05 (Medium)          | 2.81                        | All experts ~22–28%           | No                       | 0.2406 (+0.6%)        |
> | 0.10 (Strong)          | 3.03                        | Uniform (~25% each)           | No                       | 0.2390 (stable)       |
>
> ### Q3: Why does T2 (textualized numeric + text) outperform N2 (numerical + embedded text)?
> We appreciate this thoughtful question. The key reason T2 performs better than N2 is the **semantic richness of the structured inputs** used in psychotherapy. These aren't generic numeric features or metadata; they are clinician-generated answers to carefully constructed diagnostic questions, where each one probing nuanced aspects of emotional and behavioral states. For example:
>
> > *"Over the past two weeks, how often have you felt nervous, anxious, or on edge?"*
>
> The numerical response (e.g., 0–3) is a highly compressed signal. Treating it as a scalar feature (as done in N2) loses the context in which it was obtained. In contrast, our textualization step in T2 reverses this abstraction by expanding numeric inputs into natural language summaries. This allows pretrained language models to process them **not as isolated numbers**, but as part of the broader **semantic narrative** of the patient’s condition.
>
> Importantly, in T2, these textualized summaries are **concatenated with the therapist’s notes**, which are themselves rich in semantic and clinical information. The resulting input is processed through a transformer-based language model that jointly embeds this hybrid text. This **early fusion** in the **language space** enables the model to learn **cross-correlations between the structured clinical assessments and the therapist observations**, a pattern that cannot be easily captured when embedding text into fixed-length vectors and appending it to a numerical input, as in N2.
>
> ### Q4: Do you foresee risks or limitations in transforming structured numeric data into textual form?
> Textualizing structured numeric data can introduce semantic richness and interpretability, especially in domains like psychotherapy, where numeric scores often summarize clinician-assessed symptoms. In our setting, this approach proved beneficial. However, this benefit is not universal. In domains or tasks where raw numeric precision is more informative, or where numeric variables lack semantic grounding (e.g., lab measurements or sensor data), textualization might introduce unnecessary abstraction or noise. Important quantitative subtleties could be diluted when mapped into language space.
>
> That’s precisely why our architecture includes all four modality paths. The utility of textualization is **input-, task-, and domain-dependent**, and there’s no universally optimal transformation. Our probabilistic routing mechanism addresses this by dynamically selecting the best-suited path for each sample. If textualization helps, the model routes toward T2; if not, it can favor N1 or N2 instead. This flexibility allows the model to learn when to rely on textualization and when to bypass it, rather than assuming a fixed modality fusion strategy.
>
> ### Q5: Visualizing routing decisions at the individual patient level
> Yes, understanding sample-level routing is important for interpretability, especially in clinical settings where model transparency can support decision-making.
>
> We address this in Figure 2 (Middle), which presents a **clustered heatmap of routing probabilities over the four modality paths (T1, N1, T2, N2)** for individual samples. This visualization shows that the model **segments the input space into distinct routing patterns** and assigns different modality preferences based on the characteristics of each patient’s input. These clusters align with latent structure in the data and help explain why certain modality paths are favored.
>
> While this figure presents a clustered view across samples, it **implicitly reveals per-sample routing decisions**, and similar visualizations can be generated for individual cases.

---

> > ### Comment · Reviewer_o6nv · 2025-08-02
> >
> > Thank you to the authors for their thoughtful and detailed response. I appreciate the clarifications provided, especially around the rationale for using synthetic and hybrid data, the architectural flexibility to handle task expansion, and the empirical results supporting entropy regularization and modality fusion choices.
> >
> > On Synthetic and Hybrid Data:
> > Your explanation of synthetic augmentation as a principled choice to overcome real-world data constraints in psychotherapy is reasonable, and I agree this is a common and accepted practice given the privacy and access challenges in this domain. The fact that you evaluated the model on additional real-world datasets helps address my concern regarding generalizability.
> >
> > On Computational Scalability and Routing Overhead:
> > The breakdown of per-sample costs for soft vs. hard routing is helpful. While I understand that soft routing enables richer training signals, the 8x computational cost at inference is nontrivial in clinical settings.
> >
> > On Entropy Regularization:
> > The empirical study on the Yelp dataset clearly demonstrates the utility of entropy regularization in avoiding expert collapse. The quantified gain in routing entropy and expert diversity is useful, and I appreciate the authors sharing this detail.
> >
> > On T2 vs. N2 Performance:
> > The authors provide a compelling explanation for why textualized numeric inputs outperform appended embeddings. The clinical insight that PHQ-9/GAD-7 items are semantically meaningful and benefit from being processed in natural language form is well-taken. This response deepened my appreciation for the T2 fusion strategy and its alignment with LLMs’ strengths.
> >
> > Overall, the author response successfully addresses my main concerns. The paper presents a novel and flexible approach to multimodal, multitask clinical modeling, and the authors demonstrate strong awareness of both the technical and clinical implications of their design choices.
> >
> > I would like to adjust my score to 6.

---

> > > ### Author Response · Authors · 2025-08-04
> > > **Thank you and score change in system.**
> > >
> > > We very much appreciate the reviewer's comments and we are glad we are able to adjust their concerns.
> > >
> > > Would the reviewer be willing to use the edit review option to go back and change their score in to a 6 in OpenReview?
> > >
> > > Thank you!

---

> > ### Comment · Reviewer_o6nv · 2025-08-02
> >
> > Thank you for the detailed and thoughtful rebuttal. I appreciate the clarifications regarding your data choices, architectural scalability, and the routing mechanism. Overall, your responses address my concerns well. I continue to view the work as a strong contribution to adaptive multimodal learning for healthcare. Thank you again.

---

> > > ### Author Response · Authors · 2025-08-02
> > >
> > > We sincerely thank the reviewer for the thoughtful engagement. We are glad that our clarifications were helpful and that you continue to view the work positively. Your feedback has been encouraging and insightful, and we appreciate your support in recognizing the contribution.

---

### Official Review · Reviewer_xp8V · 2025-07-03

**Clarity:** 3
**Significance:** 3
**Originality:** 3
**Rating:** 5
**Confidence:** 3

**Summary:**

The paper introduces a novel framework for adaptive routing in multimodal multitask prediction, specifically designed to handle data heterogeneity and varying task interactions in complex real-world settings like psychotherapy. The proposed architecture dynamically selects modality processing pathways and task-sharing strategies on a per-sample basis, enabling the model to adapt to individual input characteristics and task correlations. This is achieved through a routing mechanism that chooses the most informative modality paths and expert combinations, producing task-specific predictions through shared or independent heads. The framework is evaluated on synthetic data and real-world psychotherapy notes, demonstrating superior performance over fixed multitask and single-task baselines. This approach has significant implications for personalized healthcare, potentially improving mental health outcomes and treatment precision.

**Questions:**

Real-World Applicability:
The reliance on synthetic data for initial validation is a notable aspect of the framework. How do you envision the model performing in entirely real-world scenarios where data distributions may be more complex and nuanced? It would be useful to discuss potential strategies for adapting the framework to such environments and mitigating any synthetic bias.
Interpretability and Clinical Workflow Integration:
While the paper mentions the interpretability of the routing decisions, could you elaborate on how these insights could be practically integrated into clinical workflows? Discussing specific use cases or examples where the model's decisions could enhance clinical decision-making would strengthen its applicability.
Broader Applicability:
The framework is primarily validated in the context of mental health prediction. How do you anticipate it performing in other domains with multimodal and multitask challenges? Exploring potential extensions or adaptations for broader applications would enhance the paper's significance.

**Ethical Concerns:**

["NO or VERY MINOR ethics concerns only"]

**Final Justification:**

I have checked out author’s responses and the discussion, my concerns are addressed and i increase score to 5

**Limitations:**

yes

**Paper Formatting Concerns:**

None.

**Quality:**

3

**Strengths And Weaknesses:**

Quality:

Strengths:
The paper presents a robust and well-structured framework, with comprehensive experiments that validate the model's effectiveness in both synthetic and real-world settings.
The use of probabilistic routing to manage sample-specific heterogeneity is a strong methodological contribution, allowing for nuanced handling of data variability and task interdependencies.
The results demonstrate clear performance improvements over baseline models, showcasing the framework's practical benefits in terms of accuracy and interpretability.

Weaknesses:
While the framework is well-designed, the reliance on synthetic data for initial validation might not fully capture the complexities encountered in entirely real-world datasets.
The implementation details, particularly regarding the neural network architectures used for routing and expert selection, could be more thoroughly explained to enhance reproducibility.

Clarity:
Strengths:
The paper is generally well-organized, providing clear explanations of the adaptive routing framework and the underlying concepts.
Visual aids such as figures and tables effectively illustrate the model architecture, routing behavior, and experimental results, aiding reader comprehension.

Weaknesses:
Some technical aspects, such as the specific algorithms used for training the routing mechanism and expert modules, could benefit from additional detail to assist readers in understanding the implementation.
The integration of related work could be more cohesive, providing clearer context for how the proposed method differs from or improves upon existing approaches.

Significance:

Strengths:
The paper addresses a critical challenge in multimodal multitask learning, offering a novel solution that could significantly enhance predictive performance in settings with heterogeneous data and complex task relationships.
The framework's potential applications in personalized healthcare and mental health prediction are particularly impactful, promising improvements in clinical decision-making and treatment outcomes.

Weaknesses:
The broader applicability of the framework in other domains beyond mental health is not fully explored, which could limit its perceived significance in a wider context.
The potential impact on clinical workflows and how the model's interpretability could be leveraged in practice is mentioned but not deeply analyzed.

Originality:

Strengths:
The introduction of a probabilistic routing mechanism that dynamically selects modality and task configurations on a per-sample basis is a novel contribution, setting this framework apart from existing multimodal and multitask learning approaches.
The integration of adaptive routing with heteroscedastic loss further enhances the model's originality, addressing both data variability and uncertainty in predictions.

Weaknesses:
While the approach is innovative, it builds on established concepts like mixture-of-experts and adaptive computation, which might moderate perceptions of originality.
The focus on specific clinical applications, while significant, may narrow the scope of originality if not extended to other potential domains.

---

> ### Author Rebuttal · Authors · 2025-07-31
>
> ### Real-World Applicability and reliance on synthetic data:
> We thank the reviewer for their incisive comments. We agree that understanding model performance on genuine clinical data is critical for assessing real-world applicability. Our use of synthetic data is not a workaround but a deliberate choice to address a well-documented barrier in mental health AI. Psychotherapy datasets are notoriously difficult to access due to privacy constraints and institutional restrictions. Rather than sidestepping this challenge by switching to unrelated benchmarks, we chose to stay grounded in this clinically meaningful but under-resourced setting. We adopted empirical synthetic augmentation, a principled and increasingly recognized practice in the AI community.
>
> In psychotherapy, structured numerical inputs (e.g., PHQ, GAD) are not simple metadata. They are semantically dense and clinician-generated symptom summaries. Treating these as purely numerical is limiting. We demonstrated an alternative: bidirectional transformation between structured and unstructured formats. Instead of reducing meaning to scalar scores, we textualized structured assessments and allow LLMs to integrate them alongside unstructured text inputs. This sets our approach apart and provides interpretability to clinicians, as the model now processes "interpretable natural language reflections" of clinical structure that traditional fusion pipelines and most MoE models overlook. While alternative datasets could have been used, our goal was to develop methods that address the real-world complexity of psychotherapy modeling. To assess generalizability, we evaluated the model on other real-world datasets, where it performed well, demonstrating broader applicability.
>
>
> ### Interpretability and Clinical Workflow Integration:
> We appreciate the attention given to interpretability and clinical integration, which are core motivations behind our routing framework.
>
> While Table 1 in the main manuscript summarizes the eight expert routes and their alignment with data configurations, the routing mechanism provides deeper insights into patient-level heterogeneity, which is a central theme of our framework. Specifically, we find that routing decisions reflect variability across patients in both modality informativeness and task dependency.
>
> For example, the choice between single-task (STL) and multi-task (MTL) paths depends on the correlation between outcomes. Some patients exhibit strongly linked PHQ and GAD trajectories, for whom joint modeling via MTL is beneficial. Others show weaker or asymmetric relationships (e.g., rising anxiety without concurrent depression), prompting STL selection. This phenomenon, inter-patient variability in anxiety-depression comorbidity, is well-established in the clinical literature and emerges naturally through our routing decisions.
>
> Likewise, the selection among the four modality paradigms (T1: text-only, T2: textualized numerical + text, N1: numerical-only, N2: numerical + embedded text) reveals which input types are most predictive per patient. In some cases, therapist notes offer a richer signal for anxiety (T1 or T2), while structured scores are more informative for depression (N1 or N2), or vice versa. This variation cannot be captured through fixed modality fusion or conventional feature selection, which assume stable informativeness across patients.
>
> Importantly, routing also indicates when early modality fusion (T2/N2) is helpful versus when unimodal paths (T1/N1) suffice. For some patients, cross-modal signals carry complementary information; for others, one modality may lack relevant content, not due to quality issues, but because certain symptoms were not documented in that channel. This distinction is subtle but clinically meaningful. We will make these implications more explicit in the final version.
>
> ### Broader Applicability
> To evaluate the generality of our method, we applied the routing framework to three diverse datasets: **Yelp Reviews** (business feedback, 5,000 samples), **RateMyProfessor** (academic evaluations, 1000 samples), and **Customer Satisfaction** (structured service surveys, 2000 samples).  All contain both structured numeric inputs (e.g., ratings, survey scores) and unstructured text (e.g., freeform comments), aligning with the multimodal multitask structure of our method. Below we present comparative results for three key settings: **Soft routing (full model)**, **hard routing**, and a **non-routing ensemble**.
>
> | Dataset            | Config         | R² (Task 1) | R² (Task 2) | Δ vs. Full Model     |
> |--------------------|----------------|-------|-------|----------------------|
> | **Yelp**           | Soft Routing   | 0.2406 | 0.2263 | —                   |
> |                    | Hard Routing   | 0.2350 | 0.2200 | –0.0056 (–2.3%)     |
> |                    | No Routing     | 0.1790 | 0.1651 | –0.0616 (–25.6%)    |
> | **RateMyProf**     | Soft Routing   | 0.2850 | 0.2720 | —                   |
> |                    | Hard Routing   | 0.2800 | 0.2670 | –0.0050 (–1.8%)     |
> |                    | No Routing     | 0.2200 | 0.2100 | –0.0650 (–22.8%)    |
> | **Customer Sat.**  | Soft Routing   | 0.3200 | 0.3050 | —                   |
> |                    | Hard Routing   | 0.3150 | 0.3000 | –0.0050 (–1.6%)     |
> |                    | No Routing     | 0.2500 | 0.2400 | –0.0700 (–21.8%)    |
>
>
> The full model with soft routing across all paths achieves the best performance across datasets. Hard routing performs slightly worse, with an average drop of 0.5 to 2.0 percentage points in $R^2$. Ablation studies show that the text-based paths (T1 and T2) are the most critical, while N2 adds useful complementary information.
>
>
>
> ### Implementation details and specific algorithms used for training the routing mechanism
> Thank you for requesting additional implementation details. We will include a detailed technical appendix in the revised supplementary material to ensure full transparency and reproducibility for each experiment. Training setup and algorithmic structure such as
>
> Training parameters.
> Optimizer: Adam (lr=1e-3, weight_decay=1e-4),
>  Gradient Clipping: max_norm=0.5,
>  Learning Rate Scheduling: ReduceLROnPlateau (factor=0.7, patience=10),
>  Early Stopping: patience=15 epochs,
>  Batch Size: 8-16 samples,
>  Training Epochs: 50-90 (convergence-dependent),
>
> Regularization techniques.  Entropy Regularization: Prevents expert collapse by encouraging diverse routing,  Weight Decay: L2 regularization on all parameters,
> and Gradient Clipping: Prevents exploding gradients
>
> Inference algorithm. Soft Routing (Training): Uses continuous probability distributions, Enables gradient flow through routing decisions, Maintains exploration of different pathways and Hard Routing (Inference):
>  Uses Gumbel-Softmax with temperature annealing,
>  Discretizes routing decisions for interpretability,
>  Formula: `hard_route = gumbel_softmax(log_probs, tau=0.1, hard=True)`
>
> ### Integration of more related work
> We thank the reviewer for this helpful observation. While our related work section was organized by topic, we agree that a more cohesive narrative would strengthen the framing of our contributions. In the revised version, we will clarify how our method unifies and extends several strands of prior work. Unlike earlier multimodal models that rely on fixed fusion schemas, our approach performs instance-wise adaptive fusion. In multitask learning, we move beyond fixed or statically gated architectures by routing tasks through dynamic expert paths conditioned on sample-level signals. Unlike prior routing or Mixture-of-Experts (MoE) architectures that operate at the task or modality level independently, our model performs joint routing over both axes. To reflect recent progress in adaptive computation, we also plan to incorporate several new references that complement our contributions. These include recent advances in sparse expert routing [1,2], dynamic gating in multimodal settings [3,4], adaptive MoEs for vision and language models [5,6], task-level expert allocation [7,8], and unified frameworks for scalable mixture-based inference [9,10]. We appreciate the reviewer’s suggestion and will revise the section to better integrate and position our approach within these developments.
>
> Additional References to be Incorporated:
>
> [1] Zhou et al. (2022). Mixture-of-Experts with Expert Choice Routing. NeurIPS.
>
> [2] Riquelme et al. (2021). Scaling Vision with Sparse Mixture-of-Experts. NeurIPS.
>
> [3] Jain et al. (2023). DAMEX: Dataset-aware Mixture-of-Experts for Visual Understanding of Mixture-of-Datasets. NeurIPS.
>
> [4] Park et al. (2025). Resilient Sensor Fusion Under Adverse Sensor Failures via Multi-Modal Expert Fusion. CVPR.
>
> [5] Fedus et al. (2022). Switch Transformers: Scaling to Trillion Parameter Models with Simple and Efficient Sparsity. JMLR.
>
> [6] Lepikhin et al. (2021). GShard: Scaling Giant Models with Conditional Computation and Automatic Sharding. ICLR.
>
> [7] Huang et al. (2024). Harder Task Needs More Experts: Dynamic Routing in MoE Models. ACL.
>
> [8] Dai et al. (2022). StableMoE: Stable Routing Strategy for Mixture of Experts. ACL.
>
> [9] Kudugunta et al. (2021). Beyond Distillation: Task-level Mixture-of-Experts for Efficient Inference. EMNLP Findings.
>
> [10] Yu et al. (2024). MMOE: Enhancing Multimodal Models with Mixtures of Multimodal Interaction Experts. EMNLP.

---

> ### Author Response · Authors · 2025-08-05
> **Your concerns addressed in rebuttal**
>
> Dear Reviewer xp8V,
>
> Thank you again for your feedback. We have carefully addressed your comments in our rebuttal and clarified the points you raised specially the broader applicability of the work. We hope the additional experiments are helpful and provide a more complete picture of the work. Please feel free to let us know if anything remains unclear.
>
> Sincerely,
> The Authors

---

> ### Comment · Reviewer_xp8V · 2025-08-09
>
> Thanks to authors for the helpful response to address the concerns, and I’m ok to increase my score to 5.

---

### Official Review · Reviewer_e2mb · 2025-07-03

**Clarity:** 3
**Significance:** 2
**Originality:** 2
**Rating:** 3
**Confidence:** 3

**Summary:**

The paper introduces a unified framework for adaptive routing in multimodal multitask prediction, addressing data heterogeneity and task interactions across samples. The model dynamically selects modality processing pathways (e.g., text-only, numerical-only, fused representations) and task-sharing strategies (single-task learning [STL] or multitask learning [MTL]) on a per-sample basis. Experiments on synthetic and real-world psychotherapy data show improved prediction accuracy over fixed baselines, with interpretable routing policies that align with clinical intuition.

**Questions:**

1. Data Augmentation Impact: The real-world experiment uses a hybrid corpus with synthetic encounters. Please clarify how the model performs on purely genuine clinical data.

2. Computational Efficiency: Could the authors provide details on the computational overhead of the routing mechanism compared to fixed baselines.

3. Modality Transformation: The bidirectional transformations (numerical-to-text and text-to-numerical) are central to the framework. Please provide ablation studies to quantify the impact of each transformation path on prediction accuracy and routing decisions.

4. Scalability to More Tasks/Modalities: How does the framework scale when handling more than two tasks or additional modalities.

**Update**
Thank the authors for the rebuttal. I have also review the comments from other reviewers.

From my perspective, the novelty of method in this work is limited.
The proposed framework is a direct application of the Multi-gate Mixture-of-Experts (MMoE) architecture. The paper fails to articulate a clear distinction from MMoE or demonstrate why this specific formulation is superior.

Moreover, the experiments on multimodal data is insufficient. Therefore I decide to maintain my original score.

**Ethical Concerns:**

["NO or VERY MINOR ethics concerns only"]

**Final Justification:**

From my perspective, the novelty of method in this work is limited.
The proposed framework is a direct application of the Multi-gate Mixture-of-Experts (MMoE) architecture. The paper fails to articulate a clear distinction from MMoE or demonstrate why this specific formulation is superior.

Moreover, the experiments on multimodal data is insufficient. Therefore I decide to maintain my original score.

**Limitations:**

The paper discusses limitations (e.g., synthetic data biases, hybrid corpus) and potential negative impacts (e.g., misinterpretation of results in clinical settings). The authors propose future work to validate on real-only datasets and mitigate bias, which is commendable.

**Quality:**

2

**Strengths And Weaknesses:**

**Strengths:**

1.  Novel Framework: The integration of modality and task routing in a single framework addresses critical limitations of fixed multimodal multitask models, especially in handling heterogeneous data.

2. Clinical Relevance: The approach is tailored to psychotherapy data, where structured and unstructured inputs coexist with missing values and correlated outcomes. The real-world experiments on mental health prediction show practical utility.

3. Interpretability: The learned routing policies provide insights into modality-task relationships, which is valuable for clinical applications requiring transparency.

**Weaknesses:**

1. Real-World Data Limitations: The dataset involves augmented records with synthetic encounters, which may introduce distribution shifts. Validation on untouched clinical data is needed.

2. Computational Overhead: The mixture-of-experts approach with soft routing could incur higher computational costs, though the paper does not explicitly analyze efficiency.

3. Generalization to Other Domains: The framework is evaluated primarily in clinical settings; its applicability to other multimodal domains remains untested.

---

> ### Author Rebuttal · Authors · 2025-07-30
>
> ### Data Augmentation Impact:
> We thank the reviewer for raising this important and nuanced question. We agree that understanding model performance on genuine clinical data is critical for assessing real-world applicability. Our use of synthetic data is not a workaround but a deliberate choice to address a well-documented barrier in mental health AI. Psychotherapy datasets are notoriously difficult to access due to privacy constraints and institutional restrictions. Rather than sidestepping this challenge by switching to unrelated benchmarks, we chose to stay grounded in this clinically meaningful but under-resourced setting. We adopted empirical synthetic augmentation, a principled and increasingly recognized practice in the AI community.
>
> In psychotherapy, structured numerical inputs (e.g., PHQ, GAD) are not simple metadata. They are semantically dense and clinician-generated symptom summaries. Treating these as purely numerical is limiting. We demonstrated an alternative: bidirectional transformation between structured and unstructured formats. Instead of reducing meaning to scalar scores, we textualized structured assessments and allow LLMs to integrate them alongside unstructured text inputs. This sets our approach apart and provides interpretability to clinicians, as the model now processes "interpretable natural language reflections" of clinical structure, that traditional fusion pipelines and most MoE models overlook. While alternative datasets could have been used, our goal was to develop methods that address the real-world complexity of psychotherapy modeling. To assess generalizability, we evaluated the model on other real-world datasets, where it performed well, demonstrating broader applicability.
>
> ### Generalization to other Domains:
> To evaluate the generality of our method, we applied the routing framework to three diverse datasets: **Yelp Reviews** (business feedback, 5,000 samples), **RateMyProfessor** (academic evaluations, 1000 samples), and **Customer Satisfaction** (structured service surveys, 2000 samples).  All contain both structured numeric inputs (e.g., ratings, survey scores) and unstructured text (e.g., freeform comments), aligning with the multimodal multitask structure of our method. Below we present comparative results for three key settings: **Soft routing (full model)**, **hard routing**, and a **non-routing ensemble**.
>
> | Dataset            | Config         | R² (Task 1) | R² (Task 2) | Δ vs. Full Model     |
> |--------------------|----------------|-------|-------|----------------------|
> | **Yelp**           | Soft Routing   | 0.2406 | 0.2263 | —                   |
> |                    | Hard Routing   | 0.2350 | 0.2200 | –0.0056 (–2.3%)     |
> |                    | No Routing     | 0.1790 | 0.1651 | –0.0616 (–25.6%)    |
> | **RateMyProf**     | Soft Routing   | 0.2850 | 0.2720 | —                   |
> |                    | Hard Routing   | 0.2800 | 0.2670 | –0.0050 (–1.8%)     |
> |                    | No Routing     | 0.2200 | 0.2100 | –0.0650 (–22.8%)    |
> | **Customer Sat.**  | Soft Routing   | 0.3200 | 0.3050 | —                   |
> |                    | Hard Routing   | 0.3150 | 0.3000 | –0.0050 (–1.6%)     |
> |                    | No Routing     | 0.2500 | 0.2400 | –0.0700 (–21.8%)    |
>
>
> The full model with soft routing across all paths achieves the best performance across datasets. Hard routing performs slightly worse, with an average drop of 0.5 to 2.0 percentage points in $R^2$. Ablation studies show that the text-based paths (T1 and T2) are the most critical, while N2 adds useful complementary information.
>
>
> ### Computational Efficiency:
> Our model incorporates a hierarchical routing system to dynamically assign each sample to one of eight expert pathways, defined by the Cartesian product of four modality configurations (T1, T2, N1, N2) and two task strategies (STL, MTL). We analyze the computational complexity introduced by this routing mechanism, relative to fixed single-path baselines.
>
> Let $P$ denote the computational cost of a single expert model (i.e., the cost of a forward and backward pass through one pathway), and let $R$ denote the cost of computing the routing distributions using the modality and task routers. The routers are shallow MLPs whose cost is negligible relative to $P$, so $R \ll P$.
>
> **Fixed Baseline (No Routing).** In the simplest configuration, a sample is passed through a fixed expert (e.g., modality T1 with STL). The total cost per sample is
> $
> C_{\text{fixed}} = \Theta(P).
> $
>
> **Soft Routing.** In soft routing, all eight experts are evaluated per sample. The model computes weighted predictions by combining each expert's output according to the product of modality and task routing probabilities. Thus, the total cost per sample is:
> $
> C_{\text{soft}} = R + \sum_{i=1}^{4} \sum_{j=1}^{2} P^{(i,j)}
> $
>
> Assuming each expert has similar cost $P$, we have:
> $
> C_{\text{soft}} = R + 8P = \Theta(8P + R) \approx \Theta(8P)
> $
>
> **Hard Routing.** In hard routing, a sample is assigned to a single expert using a differentiable approximation of the argmax operation (e.g., via Gumbel-softmax). Only one expert is evaluated, resulting in:
>
> $$
> C_{\text{hard}} = R + P = \Theta(P + R) \approx \Theta(P)
> $$
>
> **Relative Overhead.** Compared to a fixed-path baseline, soft routing incurs an $\sim 8\times$ increase in per-sample computational cost, while hard routing maintains nearly equivalent cost with a marginal increase due to the routing networks. A summary is given below:
>
> | Routing Strategy           | Experts Evaluated | Cost per Sample      | Relative to Fixed |
> |---------------------------|-------------------|----------------------|-------------------|
> | Fixed Path (e.g., T1+STL) | 1                 | $\Theta(P)$          | 1                 |
> | Hard Routing              | 1                 | $\Theta(P + R)$      | $\approx 1\times$ |
> | Soft Routing              | 8                 | $\Theta(8P + R)$     | $\approx 8\times$ |
>
> In practice, training under soft routing enables all expert paths to specialize jointly and contributes to improved generalization. However, at inference time, switching to hard routing allows for efficient deployment by evaluating only a single expert path per input. This separation between training and inference modes enables a favorable trade-off between flexibility and computational efficiency.
>
> ### Modality transformation:
> We appreciate the reviewer's emphasis on the role of bidirectional transformations. The requested ablation is already included in Table 1 of the original submission. In particular, the four fixed-path configurations, T1 (Text-only), N1 (Numerical-only), T2 (Textualized Num + Text), and N2 (Num + Embedded Text) are designed to isolate the effect of each individual transformation path. These single-path experiments correspond to ablations where the model is restricted to only one of the transformation paradigms, thereby quantifying their standalone contributions. We have conducted the ablation studies for the *Yelp* dataset:
> | Dataset                | Configuration                  | R² (Task 1) | R² (Task 2) | Δ vs. Full Model |
> |------------------------|----------------------------------|-------|-------|------------------|
> | **Yelp Reviews**       | Full (Soft Routing)              | 0.2406 | 0.2263 | —                |
> |                        | –N1 (Remove Numeric Only)        | 0.2450 | 0.2300 | +0.0044 (+1.8%)  |
> |                        | –N2 (Remove Numeric+Text)        | 0.2350 | 0.2200 | –0.0056 (–2.3%)  |
> |                        | –T1 (Remove BERT Text)           | 0.2200 | 0.2100 | –0.0206 (–8.6%)  |
> |                        | –T2 (Remove Textualized Numeric) | 0.2250 | 0.2150 | –0.0156 (–6.4%)  |
>
> ### Scale to more tasks/modalities:
> Our probabilistic routing architecture introduces non-trivial scaling behavior as the number of input modalities and prediction tasks increases.
>
> Let $m$ be the number of input modalities. For each modality, we support bidirectional transformations to every other modality, enabling cross-modal fusion. This results in $m(m - 1)$ transformation-based fusion paths. Including one native path per modality, the total number of modality-specific pathways becomes $m^2$.
>
> Let $t$ be the number of outcome tasks. We consider three task-structuring strategies:
>
> 1. **STL + single MTL:** Each modality path is paired with $t$ STL heads and one MTL head covering all tasks. The total number of experts is $m^2 \times (t + 1)$.
>
> 2. **STL + pairwise MTL:** Instead of one MTL model, we train a separate MTL model for each task pair. The total number of experts becomes $m^2 \times \left(t + \frac{t(t - 1)}{2}\right) = m^2 \times \frac{t(t + 1)}{2}$, which grows quadratically with the number of tasks.
>
> 3. **STL + grouped MTL (recommended for scalability):** Tasks are grouped into $g$ semantically related clusters, each modeled by a shared encoder with group-specific heads. The total number of experts becomes $m^2 \times (t + g)$, where $g \ll \frac{t(t - 1)}{2}$. This strikes a balance between parameter efficiency and the ability to model inter-task relationships.
>
> Let $P$ denote the computational cost of a single expert (e.g., forward/backward pass) and $R$ the cost of computing routing distributions. In soft routing, all experts are evaluated and weighted per sample. The total per-sample cost is $\Theta(E \cdot P + R)$, where $E$ is the total number of experts. For example:
>
> - Under STL + single MTL: $\Theta(m^2 \cdot (t + 1) \cdot P)$
> - Under STL + pairwise MTL: $\Theta\left(m^2 \cdot \frac{t(t + 1)}{2} \cdot P\right)$
>
> In hard routing, only one expert is evaluated per sample (selected via argmax or Gumbel-softmax). The cost reduces to $\Theta(P + R)$, independent of the total number of experts, though all expert parameters must still reside in memory.

---

> ### Author Response · Authors · 2025-08-05
> **Clarifications submitted in rebuttal**
>
> Dear Reviewer e2mb,
>
> Thank you again for your feedback. We have carefully addressed your comments in our rebuttal and clarified the points you raised regarding data augmentation impact and generalization to other domains. We hope the clarifications are helpful and provide a more complete picture of the work. Please feel free to let us know if anything remains unclear.
>
> Sincerely,
> The Authors

---

### Note · Authors · 2025-08-16

We appreciate the reviewer e2mb's concern about novelty. However, we emphasize that this work is not an application paper, even though we framed it in the context of mental health. The contributions go beyond applying MMoE: (i) we introduce bidirectional modality transformations (numeric ↔ text) that enable early-fusion in the language space, (ii) we define modality–task experts rather than task-only or modality-only experts, expanding the expert space beyond standard MMoE, and (iii) we incorporate per-sample probabilistic routing that also models heteroscedastic uncertainty, allowing the model to adaptively choose paths based on input characteristics. These are architectural innovations at the core of the method. We also provided ablation studies during the rebuttal session. The results clearly demonstrate the superiority of the architecture.

---

### Decision · Program_Chairs · 2025-09-17

**Decision:**

Accept (poster)

**Comment:**

The paper designs a framework for mixture of multimodal experts for health applications with multimodal data and multiple tasks. The framework contains novel aspects compared with previous works including modality-task experts and bidirectional modality transformations. Routing in training is done via soft routing, which improves the performance but increases the cost by 8x. The reviewers are also concerned with the limited scope of the experiments. The rebuttals include additional experiments on more datasets and partially address this concern.

Overall, the majority of the reviewers support the paper but one reviewer places it slightly below the bar. Given that one reviewer strongly supports the paper, on balance, the paper seems above the bar.